# High-resolution reconstruction of a Jumbo-bacteriophage infecting capsulated bacteria using hyperbranched tail fibers

Ruochen Ouyang[1,2], Ana Rita Costa [3,4], C. Keith Cassidy[5], Aleksandra Otwinowska[6], Vera C. J. Williams[2], Agnieszka Latka[6,7], Phill J. Stansfeld [8], Zuzanna Drulis-Kawa [6], Yves Briers [7], Daniël M. Pelt [9], Stan J. J. Brouns [3,4] & Ariane Briegel [2] ✉

The *Klebsiella* jumbo myophage φKp24 displays an unusually complex arrangement of tail fibers interacting with a host cell. In this study, we combine cryo-electron microscopy methods, protein structure prediction methods, molecular simulations, microbiological and machine learning approaches to explore the capsid, tail, and tail fibers of φKp24. We determine the structure of the capsid and tail at 4.1 Å and 3.0 Å resolution. We observe the tail fibers are branched and rearranged dramatically upon cell surface attachment. This complex configuration involves fourteen putative tail fibers with depolymerase activity that provide φKp24 with the ability to infect a broad panel of capsular polysaccharide (CPS) types of *Klebsiella pneumoniae*. Our study provides structural and functional insight into how φKp24 adapts to the variable surfaces of capsulated bacterial pathogens, which is useful for the development of phage therapy approaches against pan-drug resistant *K. pneumoniae* strains.

Disease-causing bacteria pose an ever-increasing threat to human health. While many bacterial infections can be effectively cured by antibiotics, antimicrobial resistance (AMR) is increasingly resulting in ineffective treatments. A recent study reported 3.57 million deaths associated with AMR in six leading pathogens in 2019[1]. Among these is *Klebsiella pneumoniae*, a pathogen recognized by the World Health Organization as a priority 1 (critical) for the development of new antibiotics[2]. *K. pneumoniae* can cause pneumonia, urinary tract infection, bacteremia, and other infectious diseases in humans, especially in individuals with compromised immunity[3]. The vast majority of *K. pneumoniae* clinical isolates express a pronounced capsule that is generally considered an important virulence factor mediating protection from the host's immune system[4]. There are over a hundred genetically distinct capsular locus types, of which 77 well-characterized chemical structures are used in serotyping (K types)[5].

Even though multidrug-resistant *K. pneumoniae* are insensitive to standard-of-care antibiotics, they remain susceptible to bacteriophage infection. Bacteriophages, or phages for short, are viruses that infect bacteria. Klebsiella-specific phages can successfully infect and kill their natural host; however, most of these phages are typically highly strain-specific due to the variable capsular polysaccharides (CPS) of this species, which act as a primary phage receptor. Capsule-dependent

[1]MOE Key Laboratory for Nonequilibrium Synthesis and Modulation of Condensed Matter, School of Physics, Xi'an Jiaotong University, Xianning West Road 28, Xi'an 710049, China. [2]Department of Microbial Sciences, Institute of Biology, Leiden University, Sylviusweg 72, 2333 BE Leiden, The Netherlands. [3]Department of Bionanoscience, Delft University of Technology, Van der Maasweg 9, 2629 HZ Delft, The Netherlands. [4]Kavli Institute of Nanoscience, Delft, The Netherlands. [5]Department of Biochemistry, University of Oxford, Oxford, UK. [6]Department of Pathogen Biology and Immunology, University of Wroclaw, Przybyszewskiego 63-77, 51-148 Wroclaw, Poland. [7]Department of Biotechnology, Ghent University, Valentin Vaerwyckweg 1, 9000 Ghent, Belgium. [8]School of Life Sciences & Department of Chemistry, University of Warwick, Coventry CV4 7AL, UK. [9]Leiden Institute of Advanced Computer Science, Leiden University, Niels Bohrweg 1, 2333CA Leiden, The Netherlands. ✉e-mail: a.briegel@biology.leidenuniv.nl

Klebsiella phages, including phages ΦK64-1 or vB_KleM-RaK2[6,7], are equipped with tail fibers or tailspikes serving as receptor-binding proteins (RBPs) containing CPS-degrading enzyme domains (coined capsule depolymerases) that enable successful phage adsorption and infection[8]. For simplicity, we henceforth use the term "tail fiber".

Tailed bacteriophages with genomes of >200 kbp of DNA are defined as jumbo phages[9]. The most notable structural features of jumbo phages are large capsids that encapsulate their genome[10]. The recently described vB_KpM_FBKp24 (φKp24) jumbo myophage encodes at least nine tail fibers containing different depolymerase domains, which suggests an expanded host range. Genomic analysis revealed that φKp24 has very limited similarity to any other known phage[11] and therefore constitutes a new family termed *Vanleeuwenhoekviridae* (ICTV classification in progress). Furthermore, transmission electron microscopy revealed a unique complex structure of tail fibers at the baseplate. However, more detailed insight into the structure and function of this unique set of tail fibers is currently lacking.

To gain insight into the unusual structure of φKp24, we analyzed its capsid, tail, and tail fibers using different structural methods. We generated atomic models for the highly ordered phage capsid and tail using cryo-electron microscopy (cryo-EM) single-particle analysis (SPA) combined with AlphaFold2[12] protein structure predictions and molecular dynamics (MD) simulations. The data revealed unusual features of the capsid of φKp24, most notably the composition of the entire capsid by a single MCP, and the presence of a pore in the center of the hexagons. We obtained insight into the structure of the flexible and disordered tail fibers using cryo-electron tomography (cryo-ET). This analysis was aided by machine learning approaches to train a neural network to automatically track the complex tail fibers for quantitative analysis of the tomography data. Our analysis revealed a pronounced rearrangement of the tail fibers during the infection process. In addition, we tested the infectivity of φKp24 using a K-serotype collection of *K. pneumoniae* strains, showing the unusually broad panel of CPS types targeted by the tail fibers.

The amino acid sequence prediction and binding assays against a Klebsiella serotype library in combination with a structure-function analysis of the tail fibers provided insights into their hyperbranched complex organization. The combination of structural, computational, and laboratory techniques allowed us to gain in-depth insight into the structure and function of this distinct bacteriophage.

## Results

### Capsid structure of phage φKp24

To study the structure of the φKp24 capsid, we imaged the phage using cryo-EM (Supplementary Fig. 1a, data collection parameters in Supplementary Table. 1). In our dataset, we find three variants of the capsid: full capsids filled with DNA, empty capsids, and partially DNA-filled capsids. We reconstructed the full (EMD-14356, Supplementary Fig. 1b, cyan) and empty (EMD-13862, Fig. 1a) capsid structures of φKp24 with Relion[13] to 4.3 Å and 4.1 Å resolution after polishing, and Ewald sphere correction, respectively, using the "gold-standard" $FSC_{0.143}$ criterion[14] (Supplementary Fig. 2b, c). Since the empty and full-capsid reconstructions superimpose well (Supplementary Fig. 1d), we focused our analysis on the higher-resolution reconstruction of the empty capsid (4.1 Å). The empty capsid (Fig. 1a) has a diameter of 145 nm from vertex to vertex, 130 nm along the twofold symmetry axis, and follows a T = 27 ($h = 3$; $k = 3$; $T = h^2 + k^2 + hk$) triangulation symmetry with a planar outline. (Fig. 1b).

The capsid of *Pseudomonas aeruginosa* Jumbo phage phiKZ[15], and *Ralstonia solanacearum* phages φRSL1[16], and φRSL2[17], are built in the HK97-fold[18,19] with a triangulation number of $T = 27$. Similar to these phage capsids, the capsid of φKp24 is built up by a lattice of hexamers and pentamers in each facet (Fig. 1b). The hexamers and pentamers of φKp24 are made up of copies of the same major capsid protein (MCP,

gp372, accession number QQV92002). The capsid contains a total of 260 hexamers (dark green, 13 per facet) and 11 pentamers (turquoise, one vertex is occupied by the portal complex), which represent 1615 copies of the MCP (Supplementary Movie 1). The MCPs are assembled into an icosahedral shell that encloses the phage genome. In the case of φKp24, the predicted MCP is gp372 (Fig. 1d, left).

### Structure of the major capsid protein of φKp24

Gp372 contains 597 amino acids and is to our knowledge the largest MCP currently described in any phage. Its structure was predicted and modeled using AlphaFold2[12] (Fig. 1d, left). It consists of three major parts: a "triangular body" (Fig. 1c, gray triangle, and Fig. 1d, dotted box, residues 28–88, 314–597); an "arm" (Fig. 1c, blue oval, and Fig. 1d, blue, residues 89-166); and a "hand" (Fig. 1c, orange oval, and Fig. 1d, orange, residues 167–313).

While the triangular body of MCP gp372 shows no similarity at the sequence level to the MCP gp5 of phage HK97[20,21], they exhibit some structural similarity. However, there are some notable differences between the MCP of φKp24 and HK97. Gp372 of φKp24 contains a long C-terminal loop (a component of the pore) that is absent in HK97. Additionally, the N-terminal and C-terminal domains are folded differently: (1) the A domain of HK97 gp5 monomer has four central, mostly anti-parallel β-sheets and two helices[20]. In contrast, there are five anti-parallel β-sheets and four helices of different lengths in the corresponding domain of gp372 (C-terminal domain); (2) the N-terminal arm of HK97 gp5 monomer is in a mostly unstructured conformation, while the N-terminal of gp372 folds as a long helix (Supplementary Figs. 5, 6). The arm consists of two short anti-parallel β-sheets, three short helices, and one short β-sheet close to the "hand". The most prominent feature of the hand is two layers of β-sheets, with each layer containing three anti-parallel β-sheets. Additionally, there are two short helices at the tip of "hand".

To further refine the gp372 structure according to our 3D cryo-EM data and to determine putative interactions between MCPs in the native capsid assembly, we next reconstructed models of the capsid hexamer and pentamer assemblies, including the nearest neighboring hexamers (Supplementary Fig. 9) as described in detail in the Methods section. Briefly, models of the hexamer and pentamer assemblies were first constructed via rigid docking of the predicted gp372 structure and subsequently refined to our 4.1 Å map using molecular dynamics flexible fitting (MDFF)[22]. The resulting hexamer model was then rigidly docked to the corresponding regions surrounding the refined pentamer and hexamer assemblies, and the combined structures were subjected to an additional MDFF simulation. The obtained models therefore provide representative information on each unique MCP-MCP interaction within the capsid, allowing for an assessment of key residue-residue interactions likely crucial for capsid stability.

### Inter-molecular interactions within and between hexamer units

After analyzing the inter-MCP contacts of the hexamer, we assembled a list of putative strong interactions (Supplementary Fig. 9). The triangular body of the MCPs interacts primarily with the triangular bodies of the MCPs directly adjacent and within the same hexamer (Fig. 1f). These interactions include a series of complimentary salt bridges between residue R477 and E442 on the clockwise neighbor, as well as residue R583 and both D419 and D587 on the clockwise neighbor. The triangular body also interacts strongly with the hand region of its clockwise neighbor through a series of salt bridges (Fig. 1f), including K380/E220, E398/R190, R554/E313, and R558/D163. In addition to these interactions formed between MCPs within the same hexamer, the hand region also mediates strong interactions with the hand region of an MCP from an adjacent hexamer (Fig. 1f). These involve a large cluster of charged residues, including E201,

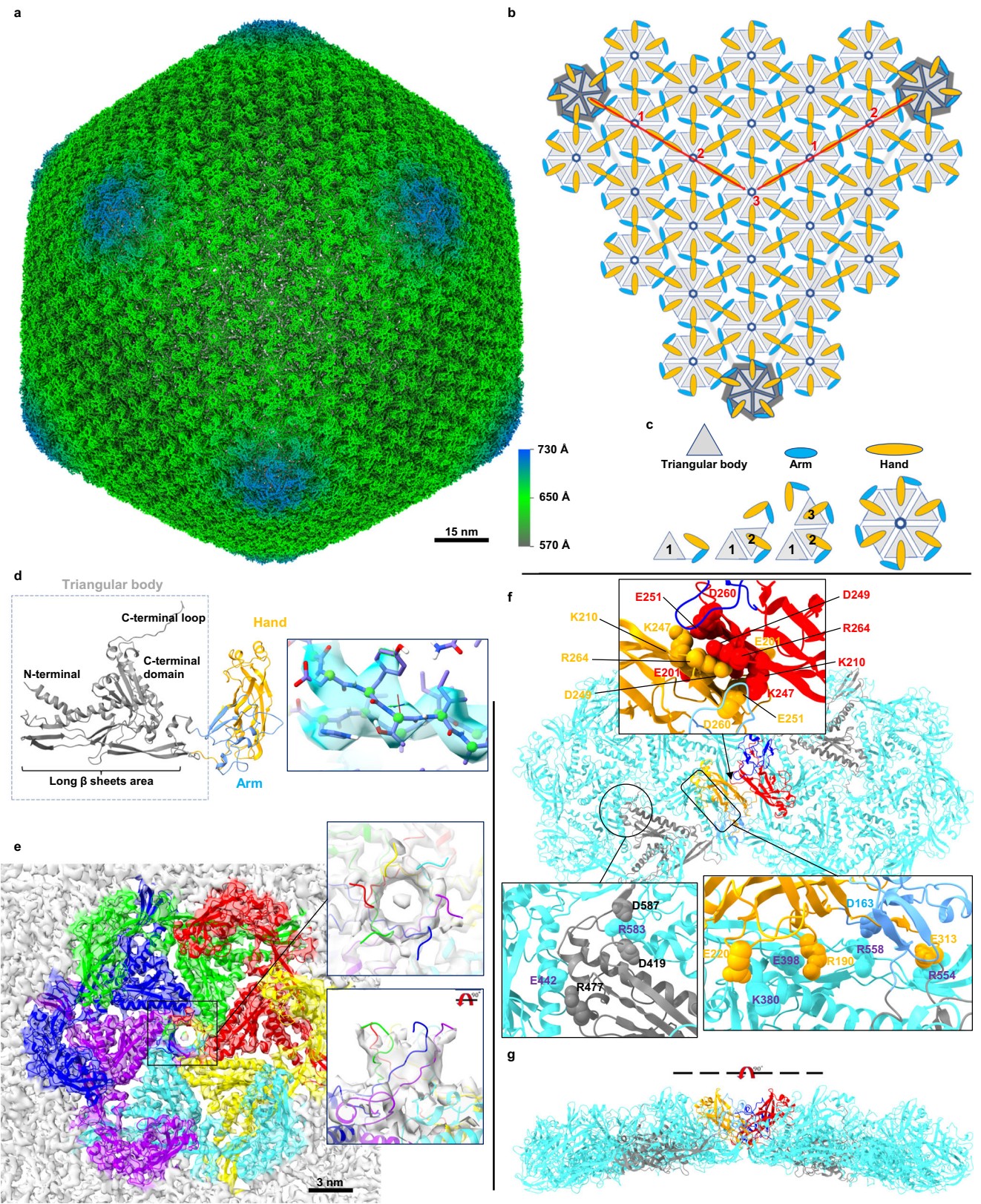

K210, K247, D249, E251, D260, and R264, which form complementary interactions. The overall conformations of the central and surrounding hexamers are quite similar, displaying a backbone root-mean-square displacement (RMSD) of ~1.5 Å. Finally, compared to other known phage capsid structures[23,24], the MCP of φKp24 displayed a structural feature that has, to our knowledge, not been previously reported: each of the six long C-terminal loops (residues Q582-S597) participate in the formation of a central helical pore with an outer diameter of ~2 nm (Fig. 1e, right upper and lower panels). Analysis of the electrostatics along the pore (Supplementary Fig. 10) show that near the entrance (from the outside into the inner capsid) it is hydrophilic, while from the opposite side, it is lipophilic. Moreover, at near the pore center, it contains an annular section that is negatively charged.

**Fig. 1 | Reconstruction and organization of the empty capsid of phage ϕKp24, hexamer arrangement, and major capsid protein. a** The dsDNA-empty capsid (EMD-13862) of phage ϕKp24 was reconstructed at 4.1 Å resolution using 19,193 particles. The displayed structure is colored radially based on the distance from the capsid center using the color bar on the lower right. **b** Schematic illustrating the ϕKp24 capsid organization. The capsid follows a *T* = 27 symmetry. The h and k volumes in triangulation symmetry calculation are highlighted in red (https://viralzone.expasy.org/8577). **c** Organization of one hexamer. A hexamer can interact with other hexamers or a pentamer via arms (blue) or hands (orange) which likely have a stabilizing function similar to decoration proteins present in other phage structures. **d** Ribbon diagram of gp372 from phage ϕKp24 generated by Alpha-Fold2. The model consists of three parts colored corresponding to the monomer of a hexamer in **c**. The right panel shows gp372 fitted into the density map. **e** Top view of an hexameric capsomere. The six monomers are colored individually. At the center of the hexamer, a pore is formed by six long C-terminal loops twisting together (enlarged rectangular box, right upper panel). The right lower panel shows the pore after rotation of the upper image by 90°. **f** Two hexamers with highlighted inter-capsomere interactions. Two gp372 hand regions interact with each other via a large cluster of charged residues, including E201, K210, K247, D249, E251, D260, and R264 (enlarged rectangular box, upper panel). The MCP triangular bodies within the same hexamer interact using complementary salt bridges between residue R477 and E442 of the clockwise neighbor, and residue R583 and both D419 and D587 of the clockwise neighbor (enlarged rectangular box, lower left panel). The triangular body also interacts with the hand region through a series of salt bridges, including K380/E220, E398/R190, R554/E313, and R558/D163 (enlarged rectangular box, lower right panel). Hydrogen bonds were analyzed using ChimeraX with relaxed distance and angle criteria (0.4 Å and 20° tolerance, respectively). **g** Side view of hexamer-hexamer interactions after rotation of the upper image by 90°.

## Pentamer arrangement, and interactions between pentamer and hexamer units

Each pentamer consists of five MCP monomers (Fig. 2a). In comparison with a gp372 in a hexamer, the N-terminal in the pentamer's gp372 shows a different folding direction (Fig. 2b, left panel). In addition, gp372 in a pentamer changes the relative orientation of the triangular body and the hand compared to the MCP in a hexagon, which makes gp372 more curved in a pentamer. (Fig. 2b, right panel). Comparison of the hexamers surrounding the pentamer with those from the hexametric assembly show that they are also slightly more curved and display an RMSD of 3–5 Å.

Overall, the MCP interaction regions in a pentamer are similar to those seen in the hexamer (Fig. 1f, Supplementary Fig. 9). However, the increased curvature of the pentamer (Figs. 1g, 2d) alters the relative orientation between the triangular body and hand regions, leading to differential stabilizing contacts. Specifically, in the pentamer MCP, residues R501 and R583 in the triangular body interact with residues D448 and E442, respectively, in the triangular body of the clockwise neighbor. The interactions with the clockwise hand region are reduced overall, involving residues K380 and R558 interacting with residues E220 and D163, respectively. In addition, as seen in the hexamer MCP, the hand region in the pentamer MCP plays an analogous role in mediating hand-hand interactions with a neighboring hexamer through a charged patch of residues described above. In contrast to the hexamer arrangement, the C-terminal loops in the pentamer do not appear to form a large, well-defined pore in the density map. Instead, they are disordered, perhaps owing to this different arrangement to the reduced space and greater steric hindrance between the MCPs in this region (Fig. 2a). Nevertheless, we do observe a small opening at the center of the pentamer that may be capable of solvent transport.

## The helical tail structure of ϕKp24

For the helical reconstruction of the tail, 450-pixels segments from phages that contained full capsids were picked. After classification, polishing, and 3D refinement, the tail structure (EMD-14357, Fig. 3a) in its extended conformation was determined to a resolution of 3.0 Å in Relion[13] (Supplementary Fig. 11b). The length of the extended tail is 180 nm on average (measured in 2D micrographs, from the collar to the tip of the spike). Based on the density map, the diameter of the tail is 24.5 nm, and the length of the sheath monomer across the longest axis is 11 nm (Fig. 3a). Like other described contractile ejection systems and bacteriophage tails, the contractile tail sheath of ϕKp24 is assembled around the inner tube. The sheath and inner tube follow the same helical symmetry with an additional 6-fold symmetry around the tail axis. The thickness of one hexameric ring is 4 nm. Using HI3D[25], the helical rise and twist are determined at 39.03 Å and 20.89°, respectively.

## Structure of sheath protein gp118

The structure of the sheath protein (gp118, accession no. QQV92013.1) was predicted using AlphaFold2 and fitted into the ϕKp24 tail EM map using ISOLDE[26] in ChimeraX[27] (Fig. 3b). The full-length sheath protein is composed of outer and central sheath components. The outer sheath component (Fig. 3c, light blue) is located at the tip of the sheath monomer, pointing outwards. The sequence analysis by ENDscript3[28] shows that the outer sheath component is a homolog of the tail sheath protein gp29PR (PDB ID: 3SPE) from bacteriophage phiKZ (Supplementary Fig. 15). This outer sheath component is not involved in the interactions between the subunits during assembly[29]. In the contracted conformation, the outer sheath components of two layers may interact with each other through electrostatic forces. This is likely since the upward-facing surfaces of the outer sheath components contain a negatively charged patch, whereas the downward-facing surfaces display a positively charged patch (Supplementary Fig. 18).

The central sheath component (Fig. 3c, orange, and dark blue) consists of the C-terminal domain, N-terminal domain, and two long extensions. The central sheath is essential in sheath assembly and contraction. To facilitate the interactions between tail sheath proteins in the tail, the two long extensions from the central sheath (the N-terminal extension, residues M1–S21; the C-terminal extension, residues A667–G689) can connect to the C-terminal domains of two adjacent gp118 proteins located in the ring below (Fig. 3d). Additionally, the C-terminal domain can connect with two adjacent sheath proteins located in the upper ring (Fig. 3b, lower right panel). Thus, three sheath proteins from two different layers link together within a so-called ridge (Fig. 3d). The central sheath component cannot interact with each other within a ring in the same layer, and all interactions between rings are confined to the ridges (Fig. 3a, d). The atomic structures show that the sheath extensions can create a "mesh" (Fig. 3d).

## The tube structure

The tube of other contractile injection systems, such as phage T4[30], R-type pyocins[31], *Serratia entomophila* antifeeding prophage (afp)[32], and the bacterial Type VI Secretion System (T6SS)[33,34], show the ability to translocate different substrates and the nature of the substrate determines the properties of the tube's channel[30]. The tube of phage ϕKp24 shows a similar function.

## Structure of inner tube-protein gp119

The structure of the tube-protein (gp119, accession number QQV92088.1, Fig. 4a) was predicted using AlphaFold2 and flexible fitted into the ϕKp24 tail EM map. We used gp119 to build a model of the tube of phage ϕKp24. One ring-like structure consisting of six tube proteins represents an assembly unit of the inner tube of phage ϕKp24 (Fig. 4b, c). There are two most prominent features of gp119: (1) the long C-terminal extension can

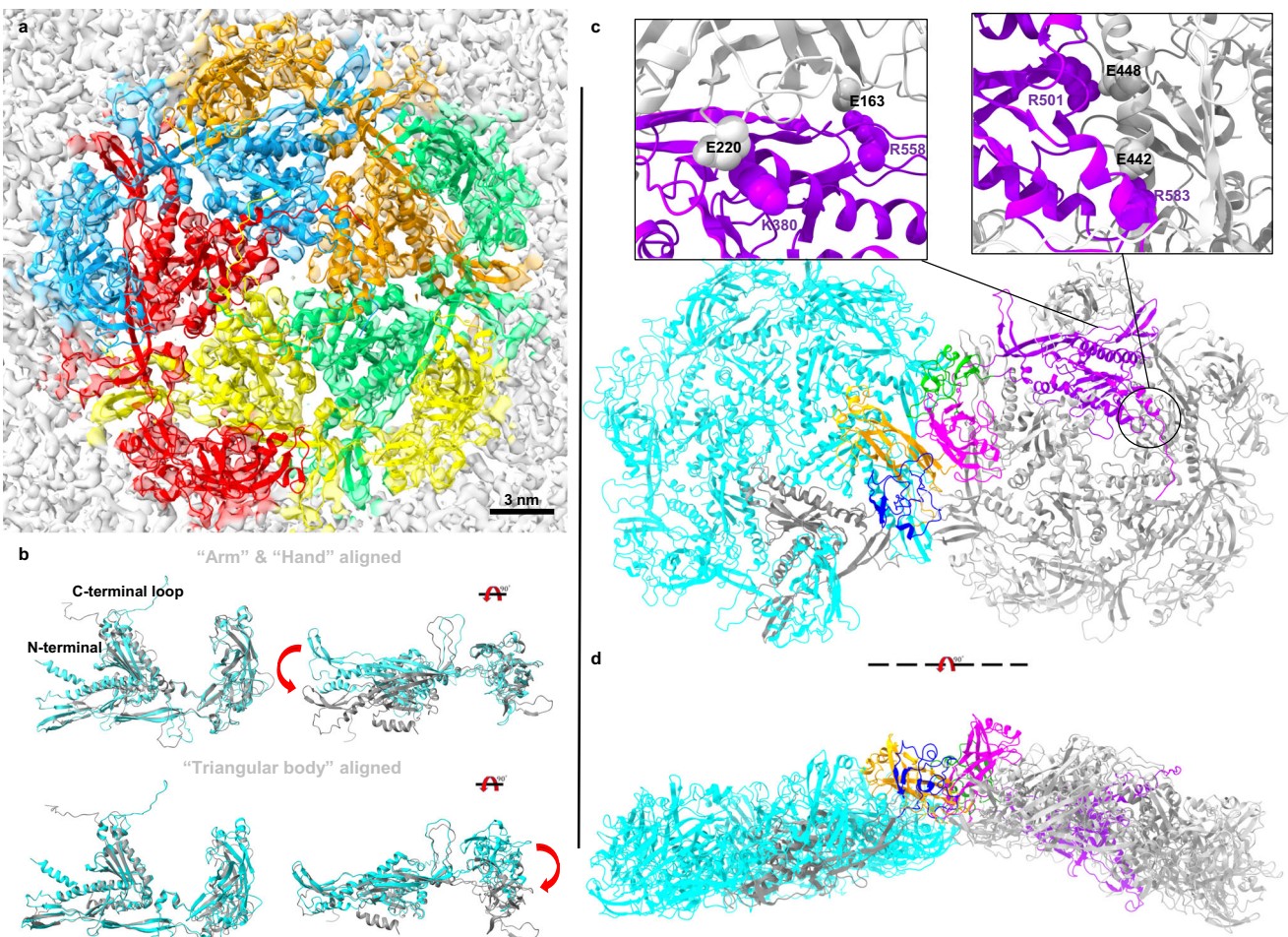

**Fig. 2 | Pentamer arrangement in the capsid of φKp24. a** Top view of a penta-meric capsomere. Five gp372 structures were fitted into the density map and colored differently. Compared with the hexamer, the pentamer lacks a structured pore at the center. Each C-terminal loop is flexible, and it tends to move to the area closer to another gp372's hand. **b** Structural comparison between the MCP (cyan) in a hexamer and MCP (gray) in a pentamer. There are some differences, such as the C-terminal loop, the N-terminal α helix, and the pendulum angle (upper red arrow) of two triangular bodies (aligned hand and arm), or the pendulum angle (lower red arrow) of two hands (aligned triangular body). The right image in b shows the comparison after rotation of the left structure by 90°. **c** Pentamer-hexamer inter-actions. The interactions between a pentamer (gray) and a neighboring hexamer

(cyan) are similar to the interactions between two hexamers. The upper left panel shows that the hand (gray) from the clockwise (CW) neighbor can interact with the triangular body (purple) in the same pentamer using salt bridges (residue E220 and K380, E163, and R558). In the pentamer MCP, residues R501 and R583 in the tri-angular body interact with residues D448 and E442 (the enlarged rectangular box, upper right panel). The gp372 from a hexamer and the gp372 from a pentamer are highlighted by different colors according to the structural units. The triangular body, arm, and hand of the gp372 in hexamer are colored black, deep blue, and orange. The triangular body, arm, and hand of the gp372 in pentamer are colored purple, green, and pink. **d** Side view of pentamer-hexamer interactions after rota-tion of the upper image by 90°.

reach another gp119 located in the layer directly above (toward the capsid); (2) the N-terminal domain can extend to a neigh-boring gp119 in a same disc (Fig. 4b). Compared the phage T4's tube-protein monomer gp19 (PDB: 5W5F), gp119 shows no sequence similarity but a similar folding at its core (gp19 has a shorter C-terminal extension and a smaller N-terminal domain) (Supplementary Fig. 21a). Both gp119 and gp19 have two layers of β sheets in common. Additionally, two long anti β sheets can extend to the neighboring tube-protein in same disc. (Supple-mentary Fig. 21). The inner surface of the tube displays a promi-nent negative charge (Fig. 4d), which prevents the negatively charged phage DNA sticks to the surface.

**Sheath-tube interaction**

Sheath-tube interactions may arise from electrostatic and non-covalent forces and from viscosity in the nanoscale gap (interstitial water) between the tail tube and the surrounding sheath[35]. The electrostatic forces are largely perpendicular to the tail tube axis and thus contribute to the connection between the sheath and the

tail tube. For phage φKp24, the sheath interacts via hydrogen bonds with inner tube: the residue R623 of the sheath protein gp118 can interact with residue D268 of tube-protein gp119 via a salt-bridge (Fig. 4e). Additionally, the sheath protein gp118 connects with inner tube subunits using electrostatic interaction (Fig. 4g). The outer surface of the tube subunits displays a positive charged patch (Fig. 4g, right panel). The C-terminal domain of the sheath protein binds to the tube's patch via a complementary negative-charged patch on one of its two α-helices.

**The model for tail assembly**

The tail tube is known to have a critical role in the assembly of the sheath in the extended state of phage tails[36]. Similar to other phage-tail systems, we propose that the assembly of φKp24's sheath starts from the baseplate, as is the case for phage T4[37]. The tube and the baseplate can form a "platform" to which the first disc of sheath subunits bind. Subsequently, the tube and the disc of extended sheath subunits serve as a scaffold for the assembly of the rest of the sheath. Thus, the assembly of the contracted state is avoided by the creation of a

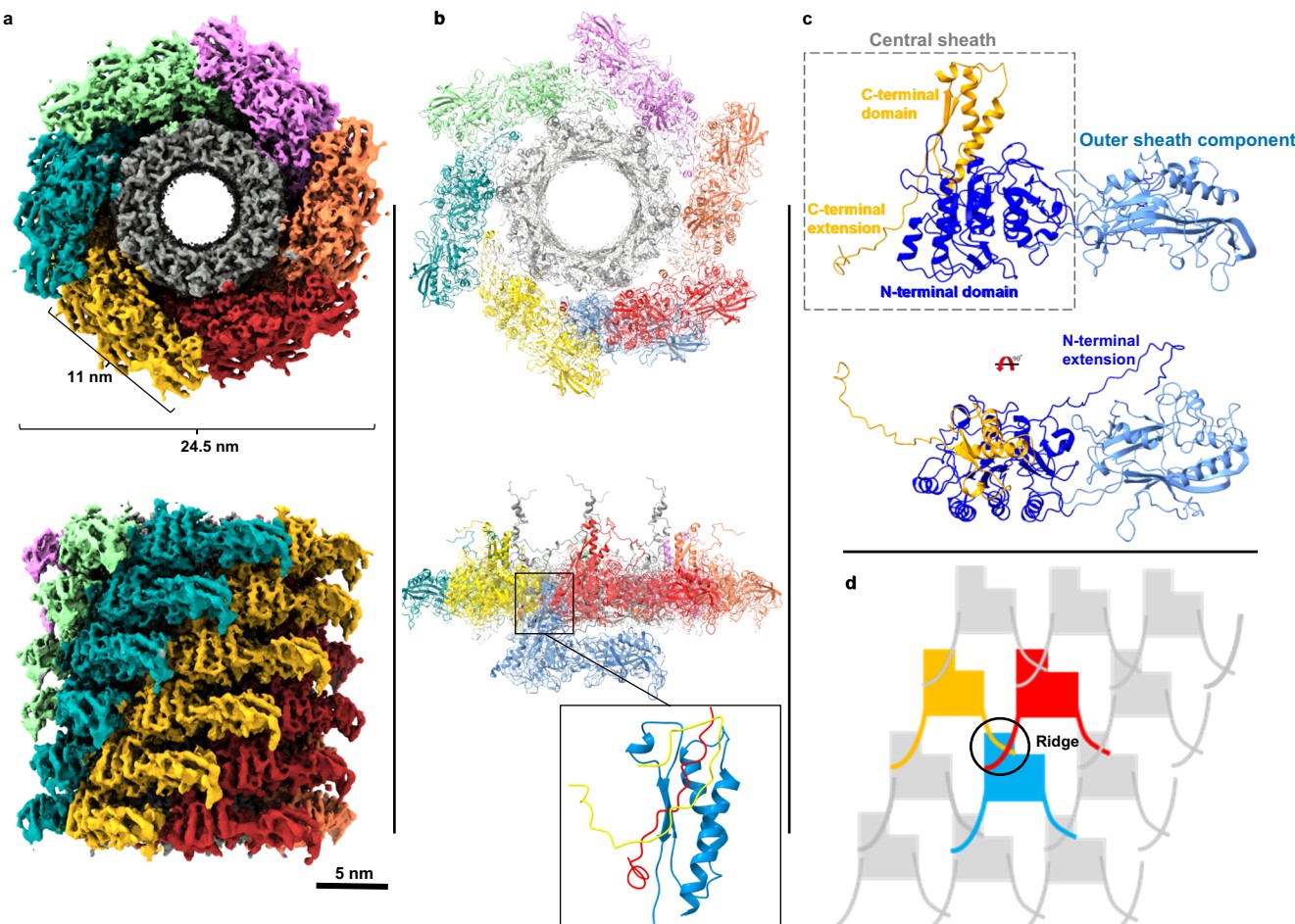

**Fig. 3 | The φKp24 helical tail. a** Top view (top image) and side view (bottom image) orientations of the 3D structure (EMD-14357) of the 3.0 Å helical tail. Each of the six helical strands is colored differently. **b** Model of tail sheath protein gp118 generated from Alphafold2. Sheath protein gp118 and the inner tube-protein gp119 were fitted into one ring of the φKp24 EM map (upper image, top view; lower image, side view). In the bottom panel, the sheath monomer (blue) that originates from the lower layer is shown to allow visualization of the ring-ring interactions. Contacts between three sheath monomers are highlighted from two successive

rings (the enlarged rectangular box, lower right panel). **c** Ribbon diagram of the predicted AlphaFold2 model of the φKp24 sheath monomer, gp118. The model consists of two parts: the central sheath (dotted box, orange and dark blue) and an extended outer sheath component (light blue). The lower image is rotated with respect to the upper image by 90° to show the N- terminal extension. **d** Schematic of the φKp24 sheath organization. The ridge, where the three sheath monomers from two layers interact, is shown in a circle.

template in which sheath subunits interact along the ridge without any lateral contacts (Fig. 3d). This template-driven assembly probably results in a metastable oligomeric structure, which could be unlocked by the baseplate for tail contraction upon interaction with the target-cell surface.

## Tomographic reconstruction and segmentation of tail fibers

To investigate the structures and the conformational changes of the tail fibers during the cell attachment and genome injection, we used cryo-ET. We imaged φKp24 together with its *K. pneumoniae* host and observed intact phages in four different states: free particles with and without DNA, and adsorbed particles with and without DNA. These four states correspond to four different states during phage infection: free phage, early-infection phages attached to the host, post-infection (empty) phages still attached to the cell, and post-infection (empty) free phages. A representative 2D cryo-ET image (Fig. 5a) shows several of these states. To find the architectures of disordered fibers at different moments during the process of infection, we focused on intact φKp24 attached to a *K. pneumoniae* cell. We selected and extracted intact phages from 89 denoised 3D reconstructions in IMOD[38].

Because of the high complexity and heterogeneous nature of the tail fibers, we chose to manually segment the tail fibers using the IMOD segmentation function[38]. We first extracted sub-volumes of individual phages from the whole tomograms (Fig. 5b). Here, we aimed at selecting phages that were in different stages of infection, from the pre- to the post-infection state. We then segmented the phage tails of these phages (Fig. 5b) and compared their overall architecture. We found that fibers of pre-infection phage are arranged around the baseplate in a near-spherical organization. In contrast, when phages attach to a host cell, the fibers interact with binding sites on the cell envelope, changing the overall architecture. The tail fibers arrange along the cell surface, creating a flat "tail plate". In addition, we observe that the tail of the attached phages is not vertically arranged with respect to the cell envelope. Instead, it is tilted, while this could be an artifact of sample preparation, it is also possible that the injection of the genome occurs at an angle in some cases.

## Tail-fiber analysis using machine learning

Due to the time-consuming manual segmentation of the tail fibers, we built a neural network[39] to auto-detect the tail fibers of phage φKp24 in reconstructed tomograms. We trained the network

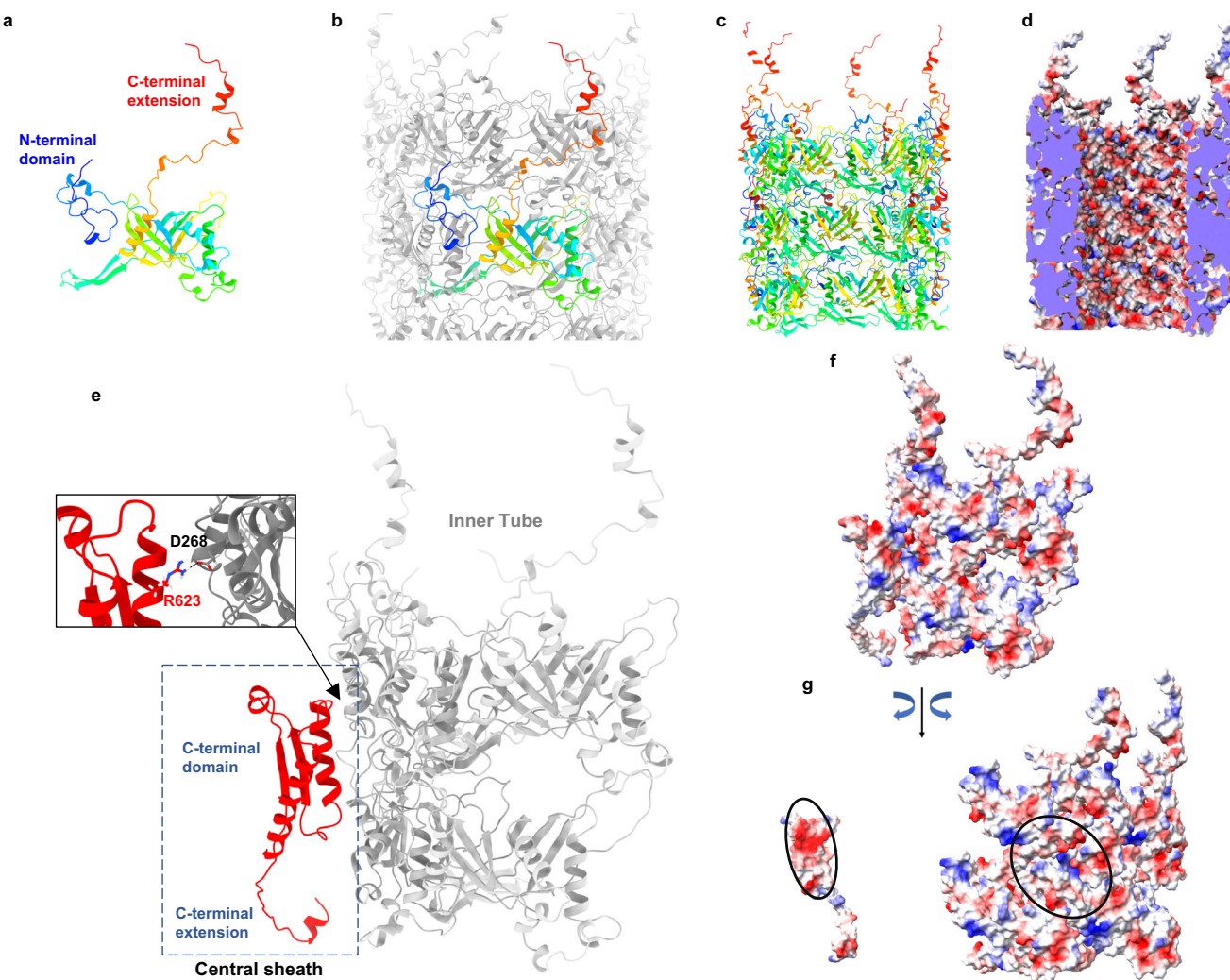

**Fig. 4 | Inner tube structure of phage φKp24 and sheath-tube interaction.**
**a** Ribbon diagram of the predicted AlphaFold2 model of φKp24's tube-protein monomer gp119, rainbow-colored in ChimeraX. **b** Ribbon diagram of two discs (hexamers) in the tube of phage φKp24, one typical gp119 was shown in rainbow color. **c** Cut-away view as ribbon diagram of the tube structure (three discs). **d** electrostatic diagrams showing the inner surface of the tube structure in **c**. Charge distribution: red = negative; blue = positive; white = neutral. Electrostatic potential was colored in ChimeraX. **e** Side views of the interface between a part of sheath protein (C-terminal domain of central sheath) and four tube-protein sub-units. The residue R623 of gp118 can interact with residue D268 of gp119 using salt-bridge (the enlarged rectangular box, up left panel). **f** The charged surface of **e**. **g** An open-book view of **f**. The complementary patches of interacting charges on both sheath and tube are marked with ovals.

using the manual segmentations of the tail fibers and, after several rounds of optimization, we applied the trained network to all tomograms, generating 3D maps containing tail-fiber structures. We define the pre-infection phages as follows: (1) the capsid is full, (2) the tail is extended, (3) the phage is intact, and (4) the phage is not in contact with a bacterial cell. Similarly, we define the post-infection phages as follows: (1) the capsid is empty, (2) the tail is contracted, and (3) there are a few connections between the tips of fibers and the cell membrane. After extraction and format conversion, 89 tail-fiber structures of pre-infection phage and 338 tail-fiber structures of post-infection attached phage were generated. 40 representative tail-fiber structures of pre-infection and tail-fiber structures of post-infection are selected by visual inspection and shown in Supplementary Figs. 24, 25. These 3D structures show clearly that each tail-fiber structure is in a different conformation. In the pre-infection phage group (Supplementary Fig. 24), the tail fibers are arranged around the baseplate in a near-spherical arrangement. The size range of the tail-fiber structures is approximately 110–130 nm in height, 150–170 nm in length, and 90–130 nm in thickness. In the post-injection phage

group (Supplementary Fig. 25), most of the tail-fiber structures are arranged in an oval, disc-like configuration. The size range in 3D is 120–150 nm in height, 160–190 nm in length, and 60–100 nm in thickness.

To further reveal the conformational changes of tail fibers in pre-infection and post-infection stages, the maps within each group were aligned to each other, and averaged to produce "canonical" fiber shapes. Finally, we got the 3D superimposed structures of tail fibers in pre-infection (Fig. 6a) and post-infection (Fig. 6b), which clearly show a conformational change in tail fiber from a near-spherical shape to a flat disc. Between these configurations, the thickness changes by approximately 40 nm. In addition, the tail from the post-infection superimposed structure is not perpendicular to the fibers' disc, which suggests that the tail is not oriented perpendicular to the host surface during DNA injection. This is also apparent in the individual structures of tail fibers and manual segmentations. We have made an animation illustrating the phage φKp24 attachment, tail fibers rearrangement and DNA ejection (Supplementary Movie 2).

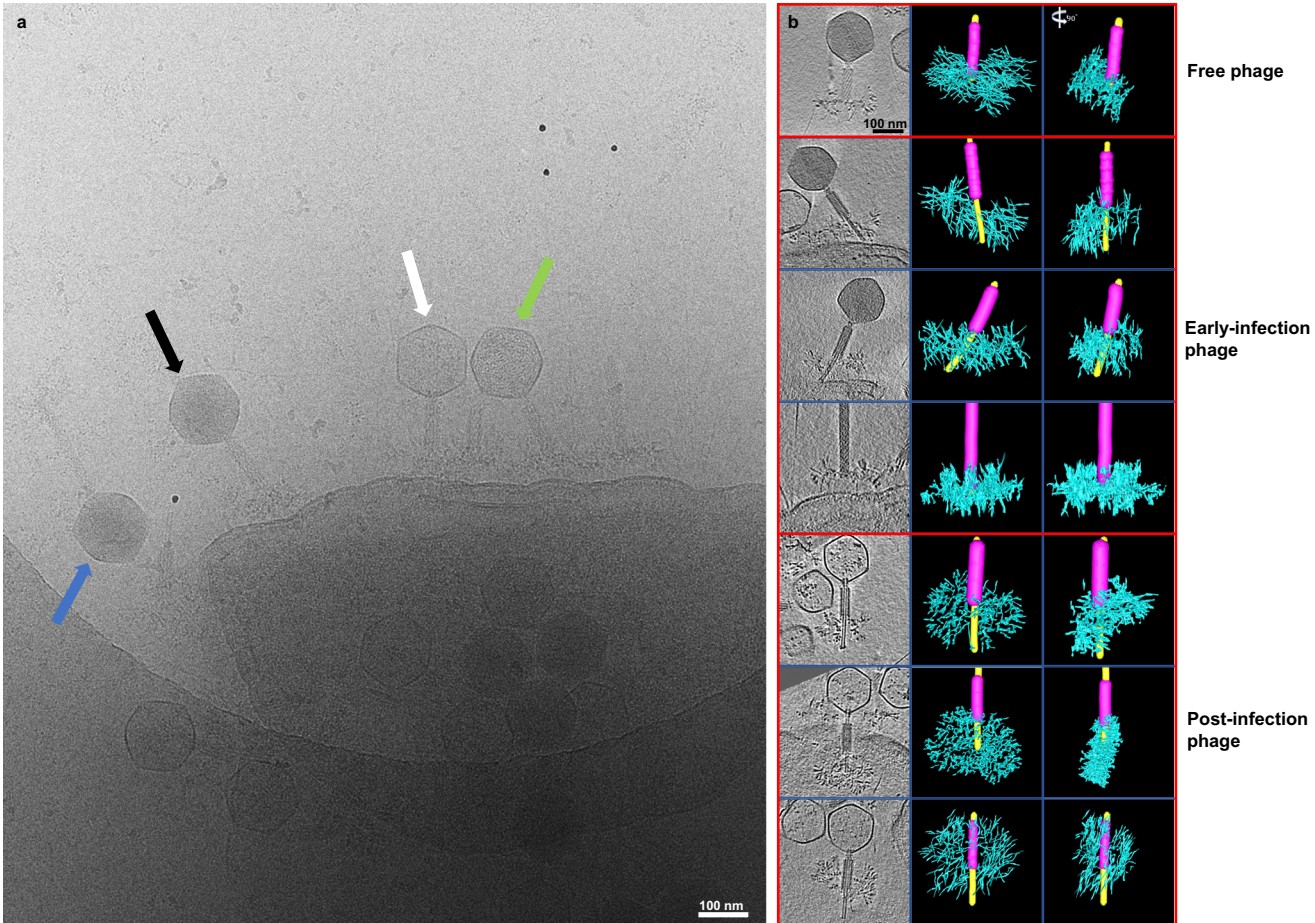

**Fig. 5 | Cryo-ET of phage φKp24 infecting *K. pneumoniae* and segmentations of tail fibers. a** 2D cryo-ET image showing several different states of phage φKp24 in its native environment: free phages (phages with dark capsids, blue arrow), early-infection phages (phages with dark capsids and attached to host membrane, black arrow), mid-infection phages (phages with half-dark capsids and attached to host membrane, green arrow), and post-infection (empty) phages still attached to the cell (white arrow). **b** Trimmed tomograms of phage φKp24 and manual segmentations. The first column shows seven representative tomograms of φKp24 in different states during the injection. From top to bottom, the first tomogram shows a free phage (full-capsid), the second to fourth tomograms show early-infection phages (full capsids) attached to a cell, and the fifth to seventh tomograms show post-infection phages (empty capsids) attached to the host. The second column shows seven manual segmentations of the tail fibers corresponding to the tomograms shown on the left. The sheath is colored purple, fibers are colored in cyan, and the tube is colored yellow. The third column shows the side view of every left segmentation after rotation by 90°.

## The complex tail-fiber architecture and relation to K-type specificities

A collection of strains with 77 different capsular (K-type) serotypes was used to characterize the infectivity of φKp24. This analysis revealed nine serotypes representative strains (K2, K13, K19, K25, K35, K46, K61, K64, and K81) susceptible to φKp24 with an efficiency of plating ranging from 1−0.001 compared to the φKp24 host strain (Supplementary Fig. 26 and Supplementary Data 1). However, sequence similarity[40] and structure-based homology modeling[41] suggest that the phage carries fourteen tail-fiber proteins (gp168, gp196, gp294, gp295, gp300, gp301, gp303, gp304, gp306, gp307, gp308, gp309, gp310, and gp313; Table 1) instead of the nine previously predicted ones[11]. These proteins possess putative depolymerizing activity (lyase or hydrolase) and a β-helical fold, which is a hallmark feature for *Klebsiella* phage depolymerases. For five of them, a putative K-serotype can be predicted based on amino acid sequence similarity to experimentally verified depolymerases (Table 1). Modeling of the fourteen proteins with RoseTTAFold[42] revealed a typical elongated protruding shape composed of parallel β-strands orthogonal to the long axis (Fig. 7, Supplementary Data 2).

Phages from Menlow group[8] (KpS110 and 0507-KN2-1), and phage ΦK64-1 group[6] (ΦK64-1 and RaK2) are the *Klebsiella* phages with the most elaborate tail-fiber apparatus described so far, comprising between five and eleven tail fibers. The N-terminus (Fig. 7, blue elements) of *Klebsiella* phage-tail fibers typically has a structural role involved in either attachment to the phage tails or to another tail fiber or providing a docking site for other tail fibers[8]. The C-terminus (Fig. 7, yellow elements) contains the enzymatic depolymerase domain defining capsule serotype specificity and features a β-helical fold. The C-terminus may also include additional chaperone or carbohydrate-binding domains[43–47]. The N- and C-terminus are typically separated by an α-helix[48] (Fig. 7, green elements). The fourteen putative depolymerases of φKp24 were divided into four groups (G1-G4) based on the length of the N-terminal structural part. The longest N-terminal part of gp306 indicates that gp306 (G1) is the primary tail fiber. In our model (Fig. 7c), its large N-terminal structural domain attaches the full hyperbranched tail-fiber system to the baseplate and serves as a docking site for secondary tail fibers (branching system) via specific T4gp10-like domains. These domains were previously experimentally confirmed in the branched tail-fiber systems of phage G7C[49] and CBA120[50]. Secondary tail fibers can comprise additional docking sites for tertiary tail fibers as described for phages belonging to the Menlow group phages[8]. As such, the four different groups correspond to more peripheral positions in the tail-fiber

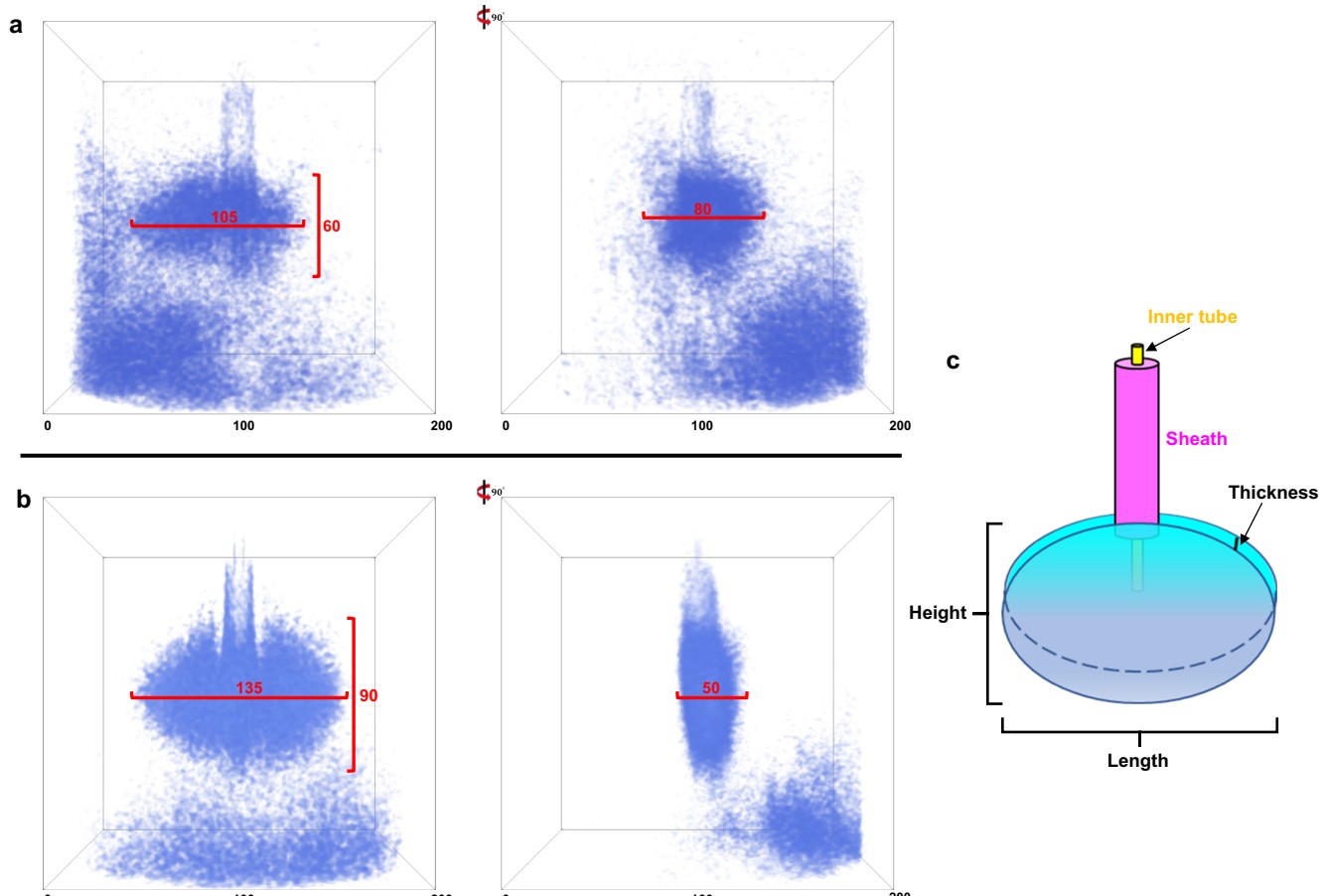

**Fig. 6 | Averaged 3D structures of tail fibers. a** 3D superimposed structure of tail fibers in pre-infection phages. 89 tail-fibers structures of pre-infection phages were aligned and averaged. The right image shows the side view of the structure after rotation by 90°. Both the outer box and red line are scale bars with pixel unit. The outer square box size is 200 pixels, pixel size is 1.312 nm. The size of the tail-fiber structure is -137.8 nm in length, 78.7 nm in height, and 104.9 nm in thickness. **b** 3D superimposed structure of tail fibers in post-infection phages. 338 tail-fiber structures of post-infection phages attached to host were aligned and averaged. The tail-fiber structure resembles a flattened disc. The size is 177.1 nm in length, 118.1 nm in height, and 65.6 nm in thickness. **c** Schematic diagram of phage sheath (purple), inner tube (yellow), and structure of tail fibers (cyan). Here we defined the height, length, and thickness of the structure.

arrangement. HHPred[51] analysis (Supplementary Data 2) detected in gp306 (G1) and gp301 (G2) the presence of a long T4gp10-like fold comprising docking sites for more than one tail fiber. Group 3 (G3) contains nine proteins (gp168, gp196, gp300, gp303, gp304, gp307, gp308, gp309 and gp310) with shorter N-terminal structure but sufficient to harbor a domain responsible for attachment to a docking site. Only one tail fiber with predicted depolymerase activity possesses a short N-terminal peptide (51 aa) preceding the separating α-helix, i.e., gp313 (G4). This peptide might mediate attachment to another tail fiber. Altogether, the tail-fiber architecture of φKp24 appears to be organized in hyperbranched peripheral layers with gp306 acting as the primary tail fiber

## Discussion

The increasing problem of antibiotic resistance in bacterial pathogens generated renewed interest in the use of phages to treat antibiotic-resistant bacterial infections. For the optimal use of phages in therapeutic applications, we need to improve our understanding of their structure and function. The recently described *Klebsiella* phage φKp24 is a promising candidate phage with an expanded host range, infecting at least nine different capsular types and likely even more based on the fourteen predicted tail fibers with depolymerase activity.

Unraveling the unique morphology represents a challenge for detailed structural studies. Especially the large size of the capsid and the heterogeneous nature of the tail fibers pose a challenge for data collection and analysis. They require the combination of different structural methods to solve the overall structure. Protein structure prediction algorithms, such as AlphaFold2 and RoseTTAFold have become invaluable tools for cryo-EM based macromolecular structure elucidation. Furthermore, machine learning provides a new way to analyze intricate image data of cryo-ET. In this study, we combined single-particle analysis and cryo-electron tomography with protein structure prediction, molecular simulations, and machine learning. This allowed us to uncover the structure of φKp24 particle, as well as characteristic morphological changes the phage undergoes during host infection.

The large size of especially the capsid together with a large amount of data also pose computational challenges during data processing. More specifically, the diameter of φKp24's capsid is around 1450 Å. According to the falling gradient of the FSC curve in Relion, the dataset used for final 3D refinement had to be binned by 1.5× to balance the final resolution with the available computational resources. The empty capsid was reconstructed to 4.1 Å resolution which is the highest resolution we can expect of the particles due to the pixel size (2.055 Å). The empty capsid of φKp24 is currently the highest resolution capsid of any jumbo phage deposited in the EMDB (the Electron Microscopy Data Bank).

The main structural component of the capsid is the MCP gp372. This unusually large protein is the basis for the hexamers and

**Table. 1 | Predicted K-serotype specificity and enzymatic activity of putative depolymerases of phage ΦKp24**

| Potential depolymerases | Potential serotype specificity | ᵃPredicted enzymatic activity | ᵇSimilar to | Accession number | Alignment with depolymerases of φKp24 | | | |
|---|---|---|---|---|---|---|---|---|
| | | | | | Cover | E value | Percent identity | Identity range |
| gp168 | - | pectate lyase, glycosidase | - | - | - | - | - | - |
| gp196 | K2 | pectate lyase, hydrolase | gp43 KLPN1 | YP_009195383.1 | 85% | 3E-176 | 50,18% | 371/570 |
| gp294 | - | xylosidase, glycoside hydrolase | - | - | - | - | - | - |
| gp295 | K1/K2/K57 | xylosidase, pectin lyase-like | gp40 Henu1 | YP_009818434.1 | 70% | 4E-54 | 37% | 152/494 |
| gp300 | K25 | pectate lyase, hydrolase | gp59 K64-1 | YP_009153199.1 | 99% | 1E-153 | 42,38% | 370/604 |
| gp301 | K35 | pectate lyase, endo-N-acetylneuraminidases | gp60 K64-1 | YP_009153200.1 | 70% | 1E-105 | 33,03% | 344/651 |
| gp303 | ND | pectate lyase, hydrolase | gp57 KpV48 | YP_009787613.1 | 97% | 8E-158 | 41,54% | 275/662 |
| gp304 | - | pectate lyase, hydrolase | - | - | - | - | - | - |
| gp306 | K1 | pectate lyase, sugar binding protein, hydrolase | gp46 KpV41 | YP_009188788.1 | 74% | 3E-144 | 40,44% | 381/680 |
| gp307 | ND | pectate lyase, glycoside hydrolase, lyase, sugar binding protein | gp45 phiKpS2 | YP_009792400.1 | 78% | 3E-75 | 34,53% | 202/585 |
| gp308 | - | pectate lyase, hydrolase | - | - | - | - | - | - |
| gp309 | KN4 | sugar binding protein, hydrolase | gp18 KN4-1 | YP_009818003.1 | 84% | 2E-126 | 38,81% | 248/639 |
| gp310 | - | pectate lyase, sugar binding protein, hydrolase | - | - | - | - | - | - |
| gp313 | - | xylosidase, glycoside hydrolase, pectate lyase | - | - | - | - | - | - |

ᵃThe predicted enzymatic activities of putative depolymerases were obtained using protein homology detection tools HMMER, HHPred, and Phyre2. *ND* no data available; no significant similarities were found.
ᵇPairwise alignment was performed on all potential φKp24 depolymerases using BlastP and alignment metrics with the best hit are shown.

pentamers that form the capsid. Typically, phage capsids are composed of one (or two) MCPs and decoration proteins (additional external proteins that link the MCPs together). Our reconstruction of φKp24's capsid has shown that the entire capsid of φKp24 can be assembled by a single MCP alone.

In the hexamers, the six long C-terminal loops form a pore in the center of a hexamer. To our knowledge, a similar pore has not been reported before in other phage capsids. The exact function of this pore is currently unclear. However, pores have been reported in viruses, for example in HIV-1. This virion uses dynamic capsid pores to import nucleotides and fuel encapsulated DNA synthesis[52]. We speculate that the pore of φKp24 may be involved in balancing pressure differences during loading of the DNA into the capsid, or during DNA ejection when capsid size increases.

There is an evolutionary link emerging between some phage-tail proteins and the Gram-negative bacterial type VI secretion system (T6SS) which is implicated in various virulence-related processes[53–55]. The structure of the contractile sheath reported here shows strong similarities to other contractile systems, such as the R-type pyocin[31] and T4[30]. This suggests that the helical tail of phage φKp24 functions in a similar fashion. Furthermore, the structure solved here suggests that the subunit-connecting extensions (C-terminal extension) of the sheath protein gp118 function as hinges during contraction, with the central sheath of gp118 likely maintaining its structure. The rearrangement of sheath subunits upon contraction leads to a much more densely packed structure. Here, the ridges are brought closer together to become partly interdigitated. In the contracted tail, the connections with the inner tube are lost, allowing the tube to insert into the target-cell membrane during infection[31].

The tomograms reveal that the tails of attached phages do not appear vertical in respect to the cell envelope. Instead, the tails of phages with full or empty capsids are tilted. While we cannot rule out that this observation is an artifact from the sample preparation, this attachment may be physiologically relevant. A tilted tail may reduce

the required energy for contraction and DNA ejection, as well as the damage to the cell envelope to prevent premature lysis of the host.

Unlike the capsid and tail, the tail fibers are structurally highly heterogeneous and not amenable for single-particle analysis. By using a neural network that is specifically designed for accurate training with a limited number of examples[56], we were able to train a network using the segmentations of the tail fibers of only seven phage particles, thereby limiting labor-intensive manual segmentation. After training, the network-enabled automatic analysis of the tail fibers of more than 600 phages, demonstrating the possibilities of machine learning for aiding the analysis of complex structures.

The tail-fiber data revealed the exquisite complexity of these tail fibers and a significant change in their arrangement between free phages and those bound to a host cell. While the fibers arrange as a sphere around the tail tip of a free phage, in the host-bound phages the tail fibers adjust to the host envelope and form a plate. Notably, the change in the conformation of the tail fibers precedes DNA release. Therefore, it may be part of the triggering mechanism itself.

Host range analysis of φKp24 revealed an unprecedented number of capsule serotypes targeted by this phage. This observation corresponds to the multiple tail fibers with specific depolymerase activity predicted in the large phage genome and the highly branched tail-fiber system that could be visualized. Structural analysis of the tail fibers indicates an extensive structural architecture to integrate all tail fibers in the phage particle with gp306 emerging as the primary tail-fiber candidate making direct contact with the phage tail, while subsequent groups of tail fibers have an increasingly shortened structural domain, hinting at a more peripheral location.

In summary, the insights gained from this novel combination of diverse methods provide an essential basis for the development for potential clinical applications. Recently, phage therapy has been successfully applied to treat an infection caused by a pan-drug resistant *K. pneumoniae* strain[13]. The extended host range of φKp24 determined here may make it an attractive candidate for the development of phage therapy.

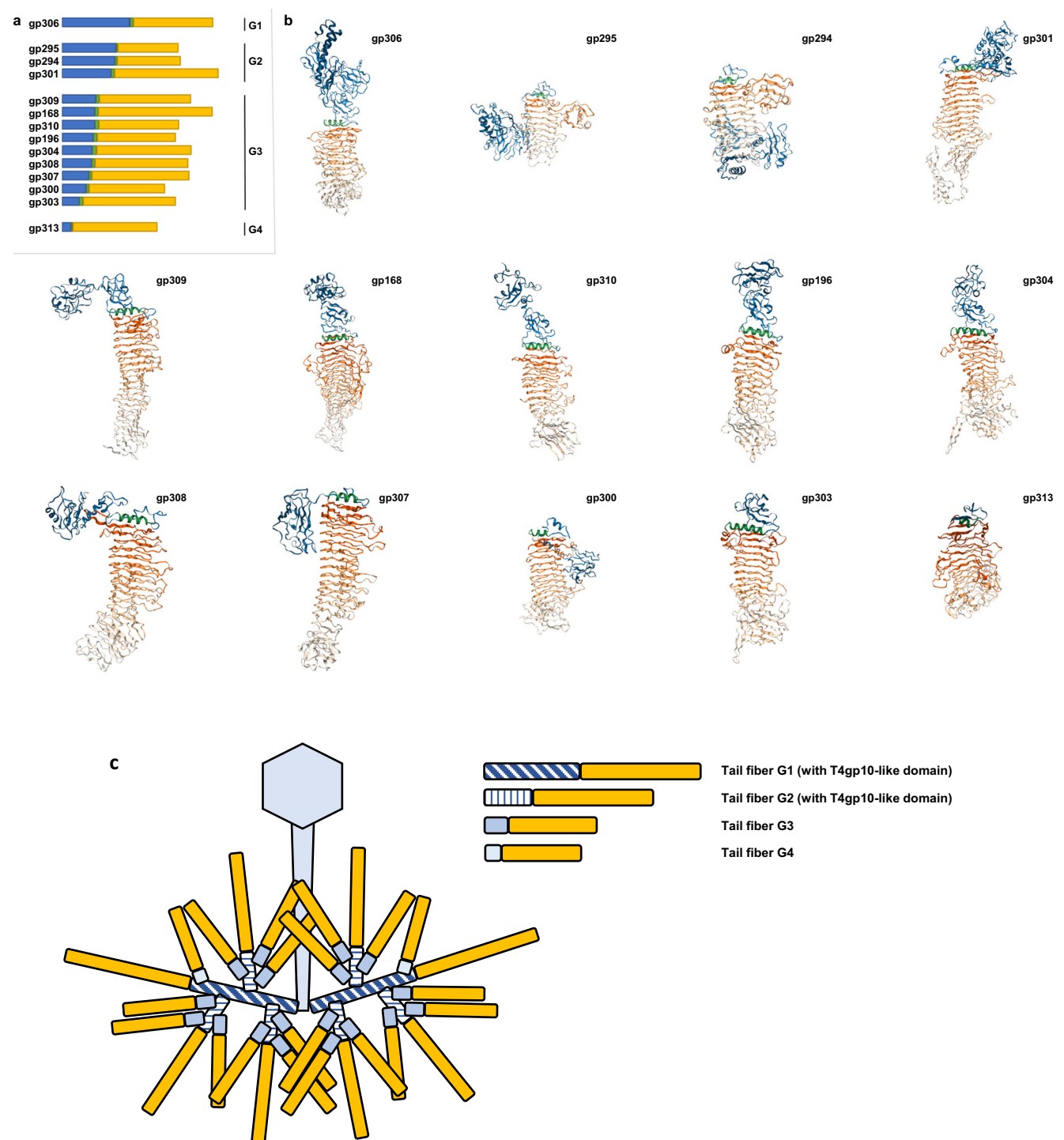

**Fig. 7 | In silico structural analyses of tail fibers with predicted depolymerase activity. a** Schematic delineation of the different modules in 14 predicted tail fibers results in four groups (G1, G2, G3, and G4) according to the length of their structural modules. Note that the relative orientations of the individual domains in the predicted structure can be different than depicted. **b** Fourteen predicted tail-fiber structures. Tail fibers with depolymerase activity are typically composed of structural modules responsible for tail attachment and branching (blue), a separating α-helix (green), and a β-helix with enzymatic activity (yellow). **c** A hyperbranching model of tail fibers of φKp24.

## Methods

### Bacterial strains

The *K. pneumoniae* reference strains used in this study (Supplementary Data 1) were obtained from the Collection de l'Institut Pasteur (CIP, Paris, France, CIP labeled strains) or purchased from The National Collection of Type Cultures (NCTC labeled strains). Bacteria were stored at −70 °C in Trypticase Soy Broth (TSB, Becton Dickinson and Company, Cockeysville, MD, USA) supplemented with 20% glycerol.

### Bacteriophage

Phage φKp24 (Genbank accession no. MW394391) was previously isolated from sewage water[11]. φKp24 was cultured in 2 L of Lysogeny Broth (LB) using *K. pneumoniae* strain K5962 as the host (Genbank

Bioproject accession no. PRJNA745534). The resulting culture was centrifuged (9000×*g*, 15 min) and the phage-containing supernatant was filter-sterilized. The phage lysate was then washed with SM buffer (100 mM NaCl, 8 mM MgSO$_4$, 50 mM Tris·HCl pH 7.5) and 10× concentrated using a tangential flow cassette (100 kDa PES Vivaflow 200, Sartorius, Germany). Phage titer was determined by spotting serial dilutions of the phage solution on double-layer agar plates of the K5962 strain for the detection of phages.

## Phage K-serotype specificity
The spot method was used to determine the serotype specificity and potential production of capsule targeting depolymerases by phage φKp24. The analyses were performed on the *K. pneumoniae* serotype collection (Supplementary Data 1, 2). The overnight bacterial cultures were suspended in fresh TSB, incubated for 2 hours at 37 °C with agitation (140 rpm) and were poured onto Trypticase Soy Agar (TSA, Becton Dickinson and Company, Cockeysville, MD, USA) plates. After drying, 10 μL of tenfold serial dilutions of phage φKp24 (10$^9$ PFU/mL as starting concentration) were spotted on bacterial lawns. After overnight incubation at 37 °C, lawns were inspected for plaque and halo zones. The efficiency of plating was calculated by dividing the titer of the phage at the terminal dilution on the test strain by the titer of the same phage on its host strain.

## Bioinformatic analysis and structural modeling of putative tail-fiber genes
The genomic sequence of the phage φKp24 was obtained from the GenBank database (NCBI Accession No MW394391). All proteins of phage φKp24 were analyzed with Phyre2[41] in search of putative tail fibers with depolymerase activity. Criteria for the prediction of putative depolymerase activity were as in[8] with some modifications: (1) the protein is longer than 200 residues; (2) the protein is annotated as tail/tail-fiber/tailspike/hypothetical protein in the NCBI database; (3) the protein shows homology to domains annotated as lyase [hyaluronate lyases (hyaluronidases), pectin/pectate lyases, alginate lyases, K5 lyases] or hydrolase (sialidases, rhamnosidases, levanases, dextranases, xylanases, glucosidase, galacturonase, galacturonosidase, glucanase) with a confidence of at least 40% in Phyre2; (4) the length of homology with one of these enzymatic domains should span at least 100 residues; (5) a typical β-helical structure should be predicted by Phyre2. Predicted proteins were modeled with RoseTTAFold[42]. The predicted models were further analyzed and the last α-helix before the β-helical structure was chosen as an ending point for the N-terminal structural part. Proteins were ranked by the length of the N-terminal structural part, starting with the longest one. The proteins were further analyzed with BlastP[40] and HMMER[57]. BlastP was used to determine the serotype specificity based on the amino acid sequence similarities of the depolymerases found in the database. The following criteria were determined for the analysis: (i) the homologous protein should have a confirmed depolymerase activity and specificity for the particular *K. pneumoniae* serotype, (ii) the amino acid sequence identity should exceed 30%, and (iii) the length of the homology should be more than 100 amino acids. The N-terminus of each selected protein was analyzed with HHPred[51,58] using pairwise alignment in search of T4gp10-like domains. Default parameters were used: MSA generation method: HHblits = >UniRef30; MSA generation iterations: 3; E-value cutoff for MSA generation: 1e-3; Min seq identity of MSA hits with query (%): 0; Min coverage of MSA hits (%): 20; Secondary structure scoring: during_alignment; Alignment Mode:Realign with MAC: local:norealign; MAC realignment threshold: 0.3; Max target hits: 250; Min probability in hitlist (%): 20. The alignment was performed against T4gp10 (NP_049768.1), the N-terminus of CBA120gp163 (orf 211, YP_004957865.1), CBA120gp165 (orf 213, YP_004957867.1) and G7Cgp66 (YP_004782196.1).

## Sample preparation for cryo-ET
*K. pneumoniae* K5962 was grown overnight at 30°C with shaking at 180 rpm. The overnight culture was used to inoculate 10 mL of fresh LB media and grown for 2 hours at 30 °C, 180 rpm. Subsequently, 1.6 mL of phage φKp24 (9 × 10$^{12}$ PFU/mL) were added to the culture and incubated for an additional 80 min. The culture was visually checked to confirm cell lysis, and 200 μl were centrifuged (5000×*g*, 5 min) and the pellet was re-suspended in the same volume of fresh media. Finally, 5 μl of 10-nm-sized gold beads (Cell Microscopy Core, Utrecht University, Utrecht, The Netherlands) were added to 50 μl of the prepared bacterial-phage mixture. Using the Leica EM GP (Leica Microsystems, Wetzlar, Germany), 3.8 μl of the mixture were applied to a glow discharged Quantifoil R2/2, 200 mesh Cu grid (Quantifoil Micro Tools GmbH, Jena, Germany) and incubated 30 s before blotting at 20 °C with ~95% relative humidity. The bacterial-phage grids were blotted for 1 s and automatically plunged into liquid ethane. Vitrified samples were transferred to storage boxes and stored in liquid nitrogen until use.

## Sample preparation for SPA
φKp24 was concentrated as follows: 1000 μl phage φKp24 liquid (9 × 10$^{12}$ PFU/mL) was spun down at a lower speed (5 k) for 2–3 times (15 min/per spin) until buffer stopped eluting from a protein concentrator (10 kDa cellulose filter). Subsequently, the sample was immediately used for plunge freezing. Using the Leica EM GP, 3.5 μl of the concentrated phage was added to a glow discharged Quantifoil R2/2, 200 mesh Cu grid, incubated 10 s at 18 °C with ~95% relative humidity. The phage grids were blotted for 0.7 s and automatically plunged into liquid ethane. Vitrified samples were stored in liquid nitrogen until use.

## Imaging conditions of Cryo-ET
The grids containing *K. pneumoniae* K5962 and φKp24 were clipped and loaded into a Titan Krios (Thermo Fisher Scientific (TFS)) equipped with a K3 BioQuantum direct electron detector and an energy filter (Gatan, Inc), which was set to zero loss imaging with a slit width of 20 eV. Imaging targets were chosen by selecting bacterial cells that were in a hole of the carbon film of the EM grid. Bacteria that appeared less dense at low magnification indicated progressed phage infection. Data were collected as movie frame stacks using SerialEM set to a dose symmetric tilt scheme between −54° and 54°, with 2° tilt increments[59,60]. The selected nominal magnification was 26,000, which corresponds to a pixel size of 3.28 Å in the collected images. The defocus of the collected data ranged from −4 to −6 μm. The estimated total dose per tilt series was 100 e/Å$^2$.

## Imaging conditions of SPA
φKp24-containing grids were clipped and loaded into a Titan Krios electron microscope (Thermo Fisher Scientific, TFS) operated at 300 kV, equipped with a Gatan K3 BioQuantum direct electron detector (Gatan). Movies were recorded using EPU (Thermo Fisher Scientific, TFS) and AFIS (aberration-free image shift) in super-resolution mode at 64,000 nominal magnification, corresponding to a calibrated pixel size of 0.685 Å with a defocus range of −1 to −5 μm and a total dose of 30 e/Å$^2$(see data collection parameters in Supplementary Table. 1).

## Cryo-ET data processing
Motion correction, frame alignment, tilt series alignment using gold fiducials, and tomogram reconstruction were carried out using IMOD[38]. The datasets were binned by a factor of 2. From the reconstructed tomograms, φKp24 phages with clearly visible baseplates, and tail fibers were picked for segmentation using IMOD. Visualization of data was performed by IMOD and Fiji[61].

## SPA reconstruction of the capsid

The capsid of phage φKp24 was reconstructed using Relion 3.1.2[13]. MotionCorr[62] was used to correct beam-induced particle movement. The contrast transfer function was estimated by CTFFIND 4.1.18[63]. Initially, 61 capsid particles were manually picked for 2D classification to generate reference templates for auto-picking. The Relion auto-picking was then used to automatically pick 290,280 particles, which were binned by 4 for 2D classification during extraction. After multiple rounds of 2D classification, particles with false-positive and contaminating features were discarded resulting in a 28,090 particle dataset, that was then classified into two classes: empty capsids and full capsids. To balance the final resolution and speed of computation, the dataset was down-sampled 1.5× with a final pixel size of 2.055 Å, a particle box of 800 pixels, and a circular mask diameter of 1600 Å. The extracted particles were then subjected to a 3D initial model and classification. The class with the largest number of particles was used for final 3D refinement (This separates and removes partially DNA-filled capsids). Finally, we chose both empty capsid class and full-capsid class for 3D refinement separately using I1 icosahedral symmetry, and the Ewald sphere correction using a single side-band image processing algorithm[64] was applied after 3D refinement. Final reconstructions were sharpened and locally filtered by Relion post-processing (workflow of 3D reconstruction, Supplementary Fig. 2a). The map resolution was estimated at the 0.143 criterion of the phase-randomization- corrected FSC curve calculated between two independently refined half-maps multiplied by a soft-edged solvent mask. The maps were displayed using UCSF ChimeraX[27].

## Helical reconstruction of the tail

Helical reconstruction was carried out using RELION 3.1.2[13]. The φKp24 tail connected to a full-capsid was picked manually. A total of 4707 manually picked segments were extracted with a 450-pixel box and then 2D classified. The initial twist value was approximated from the observed crossover distance in raw micrographs and 2D class averages. The highest proportion of the classes with clear boundaries was used as a template for final auto-picking. The parameters for helical auto-picking were optimized on a subset of 741 micrographs. After auto-picking from the whole dataset, 114,132 particles were extracted and binned by 2. Multiple rounds of 2D classification were performed to remove false-positive particles and contaminating features. The remaining particles were subjected to several rounds of 3D classifications using a featureless cylinder as the initial 3D model. Three-dimensional refinement was performed, followed by a 3D classification where the particles were classified into four classes. Particles in the class with the largest number were selected, re-extracted using unbinned data, and 3D refined with C6 symmetry. The helical rise and twist parameters for final 3D refinement were analyzed using HI3D[25] from the Jiang lab (Supplementary Table. 1), the helical rise and twist are determined at 39.03 Å and 20.89°, respectively. Final reconstructions were sharpened and locally filtered using Relion post-processing (Helical reconstruction workflow, Supplementary Fig. 11). The local resolution was generated in ResMap[65] (Supplementary Fig. 12).

## Molecular modeling

The 3D structures of φKp24 proteins were inferred using Alphafold v2.0 via the ColabFold[66] notebook at: https://colab.research.google.com/github/sokrypton/ColabFold/blob/main/AlphaFold2.ipynb. Initial models of the capsid hexamer and pentamer assemblies were constructed by rigidly docking the AlphaFold2-predicted gp372 model into corresponding regions of the capsid cryo-EM map using ChimeraX. The assembled models were then refined separately using the cascade MDFF protocol[67], which conducts sequential MDFF simulations to a series of gaussian-smoothed maps of increasing resolution, ending with the experimental map. A total of 5 × 1 ns MDFF simulations

were carried out for each assembly with symmetry restraints applied to the backbone atoms of each monomer. The resulting hexamer model was then docked into the densities surrounding the hexamer and pentamer assemblies to produce extended capsid models, which were each then subjected to an additional 5-ns MDFF simulation. All simulations were conducted using NAMD v2.14[68] and the CHARMM36 force field[69]. MDFF simulations were performed in the NVT ensemble at 300 K and using generalized Born implicit solvent. All backbone atoms were fit to the density map using a coupling constant of 0.3 with additional harmonic restraints applied to prevent loss of secondary structure, chirality errors, and the formation of cis-peptide bonds. Additional parameters were taken as the defaults provided by the MDFF plugin in VMD[70].

The AlphaFold2-predicted gp118 structure was fitted rigidly into the tail density map using ChimeraX, and the sheath region was then flexibly fitted by interactive molecular dynamics simulation in ISOLDE[26]. The PHENIX[71] software package was used for real-space refinement. The validation including CC, RMSD from ideal bond lengths and angles was also analyzed using PHENIX (Supplementary Fig. 14, and Supplementary Table. 2). To build the model of the inner tube, we first analyzed all proteins of φKp24 using HMMER[72] to identify the tail tube-protein by homology search, since this was not annotated in the Genbank file. Next, we predicted the structure of tail tube-protein gp119 using Alphafold2 and used this as a template to build the inner tube model. The model was rigidly fitted into the corresponding regions of the tail density map in ChimeraX, optimized in ISOLDE, and refined in the PHENIX application utilizing the rigid body and real space. The CC and refinement statistics, including RMSD from ideal bond lengths and angles were analyzed by PHENIX (Supplementary Fig. 20, and Supplementary Table. 2).

## Tail-fiber analysis using machine learning

A 100-layer mixed-scale dense neural network[39] was trained to detect phage φKp24 tail fibers in reconstructed tomograms. Here, we used a similar training setup as in other studies that use mixed-scale dense networks[56,73,74]. For more information about this training setup, we refer to these earlier studies. For training, manual segmentations of the tail fibers of seven phages were used, in addition to five manually selected regions without any fibers present. The ADAM algorithm[75] was used to minimize the cross-entropy loss, using random rotations and flips for data augmentation. The training was stopped after no significant improvement in the loss was observed, resulting in a training time of a few hours. Afterward, the trained network was applied to all tomograms, resulting in 3D maps of tail-fiber structures. To further analyze these maps, additional manual annotation was performed on the tomograms, indicating for each phage: (1) the center of the fiber structure, (2) the position of the head, and (3) whether the head was full, empty, or missing. Using this information, a 3D map of the fiber structure of each phage was extracted from the 3D fiber maps, resulting in individual maps for 89 phages with full heads, 338 phages with empty heads, and 181 phages with missing heads. The maps within each group were aligned to each other using both the manually annotated position information and subsequent automatic alignment by maximizing cross-correlation. The aligned maps were then averaged to produce 'canonical' fiber shapes for each group. The extracted structures and averaged structures were displayed using Paraview[76].

## Reporting summary

Further information on research design is available in the Nature Portfolio Reporting Summary linked to this article.

## Data availability

The 3D cryo-EM maps generated in this study have been deposited at EMDB (the Electron Microscopy Data Bank, https://www.ebi.ac.uk/emdb/) with accession code EMD-13862 for empty capsid of *Klebsiella*

jumbo phage φKp24, code EMD-14356 for full-capsid for *Klebsiella* jumbo phage φKp24, and code EMD-14357 for a length of the extended tail. Protein prediction data reported in this paper will be shared by the lead contact upon request. The atomic coordinates generated in this study have been deposited at wwPDB (Worldwide Protein Data Bank, http://www.wwpdb.org/) with accession PDB ID: 8BFL for hexamers of the empty capsid of Klebsiella jumbo phage φKp24, PDB ID: 8BFP for pentamer-hexamers of the empty capsid Klebsiella jumbo phage φKp24, PDB ID: 8AU1 for a length of outer sheath of the extended tail, and PDB ID: 8BFK for a length of inner tube of the extended tail.

## Code availability

All original code has been deposited at GitHub and is publicly available https://github.com/dmpelt/jumbo-bacteriophage, https://doi.org/10.5281/zenodo.7277351. Any additional information required to reanalyze the data reported in this paper is available from the lead contact upon request.

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

## Acknowledgements

This work is part of the research program National Roadmap for Large-Scale Research Infrastructure 2017–2018 with project number 184.034.014, which is financed in part by the Dutch Research Council (NWO). C.K.C. and P.J.S. are funded by BBSRC, MRC, and Wellcome. The depolymerase experiments and analyses were supported by National Science Centre, Poland, with grant numbers UMO-2017/26/M/NZ1/00233 to Z.D.-K., A.L., and UMO-2020/38/E/NZ8/00432 to A.O. A.L. holds a junior postdoctoral fellowship of the Research Foundation—Flanders (FWO) with grant number 1240021 N. D.M.P. is supported by NWO with grant number 016.Veni.192.235. S.J.J.B. is supported by the NWO with VICI grant VI.C.182.027 and the European Research Council (ERC) CoG with grant no. 101003229. R.O. was supported by the China Scholarship Council (CSC) with project number 201906280465. We thank Jamie Depelteau and Adam Sidi Mabrouk for cryo-EM sample preparation help; Wen Yang and Willem Noteborn for cryo-EM data collection; Ludovic Renault and Frédéric Bonnet for cryo-EM data processing help; Boris Estrada-Bonilla for technical support; Xiao Zhang for help with the extraction of the network results and Lei Zhang for critical feedback on the manuscript. The *K. pneumoniae* reference strains used in this study (CIP labeled strains) were obtained from the Collection de l'Institut Pasteur (CIP, Paris, France).

## Author contributions

Conceptualization: A.B., S.J.J.B, Y.B., Z.D.K.; methodology: A.B., R.O., Y.B., Z.D.K., C.K.C.; software: R.O., C.K.C; formal analysis: R.O., C.K.C, D.M.P., A.L., A.O, V.W.; investigation: R.O., A.R.C., C.K.C., A.O.;

resources: A.B., S.J.J.B, D.M.P, Y.B., Z.D.K.; data curation: R.O., A.B., Y.B., Z.D.K.; writing original draft: R.O., A.B., Y.B., D.M.P., Z.D.K.; writing— review and editing R.O., A.R.C., C.K.C., V.W., P.J.S., D.M.P., S.J.J., Y.B., A.L., Z.D.K., A.O.; visualization: R.O., V.W., Y.B.; supervision: A.B., S.J.J.B Y.B., Z.D.K.; project administration: A.B.; funding acquisition: A.B., S.J.J.B., C.K.C., P.J.S., Z.D.K., A.L., D.M.P., O.R.

## Competing interests

The authors declare no competing interests.
