## [Peer Review File · Nature Communications]

High resolution reconstruction of a Jumbo bacteriophage infecting capsulated bacteria using hyperbranched tail fibersREVIEWER COMMENTS

Reviewer #1 (Remarks to the Author):

The paper "High resolution reconstruction of a Jumbo bacteriophage infecting capsulated bacteria using hyperbranched tail fibers" by Ouyang et al. describes the structural analysis of the jumbo myophage ϕ Kp24. The structure of the head of this phage, resolved at 4.3 Å (the highest up to now deposited on the EMDB data base) reveals a triangulation number $T=27$ and that this structure is based on a single major capsid protein. The structure of the tail of this phage was also solved at 4.1Å which also allowed to generate a quasi-atomic model of this part. Finally, this work focused on the hyperbranched fibers of this phage. An electron tomography approach coupled with machine learning methods was used to show dramatic rearrangement upon attachment to the cell surface.

If several points of this manuscript are very interesting, new and the approach very elegant (the analysis of the fiber in particular), several technical points deserve to be improved and explained in more detail before publication:

Major points :

- 1- Almost no mention is made of previous structural work (even at low resolution) on jumbo phages and in particular phages with triangulation number $T=27$ (RSL1, RSL2, ϕ KZ). The same is true for phages showing multiple fibers on the baseplate or tail.
- 2- The general EM table is missing (Data collection ; Refinement ; Model composition; RMS deviations, validation, Ramachandran plot).
- 3- It is written: line 980 : The full capsid (cyan) is a little smaller than empty capsid (gray). What does "little" mean?
- 4- The phage capsid is large and therefore the defocus difference between the top of the capsid and the bottom of the capsid in the ice is important. Have you tried to perform correction of the Ewald sphere during image processing?
- 5- Comparison with HK97 MCP: is there evidence of cross-linking? The hypothesis is not even mentioned? Please add a comment on that.
- 6- Has PISA been used to see/ validate the aa interactions?
- 7- Is reference 18 the correct one? I don't think that the number of MCP is mentioned in the reference.
- 8- Comparison between the hexamers: please provide RMSD
- 9- Figure 2B : the superimposition of the hexamer MCP and the pentamer MCP have been performed on the arm part. Is it possible to also do it on the triangular body.
- 10- Tail: it seems that the picking was done in adjacent boxes. Normally there can be an overlap between boxes (it will increase the number of boxes and the final resolution) and moreover by imposing the helical symmetry and a C_6 symmetry this would greatly improve the resolution. I suggest that this be done. This would increase the resolution (better than 3.5Å) and directly enable building of the chain.
- 11- What are the helical parameters of the tail?
- 12- Since there are empty capsids, there are certainly contracted tails. Why was no analysis done on the contracted tails? It could be nice to compare the contracted and non-contracted tails.
- 13- No image analysis was performed on the baseplate. Why?
This would allow 2 things:
 - To see the origin of the fibers and maybe even solve the structure or part of the structure of some of them.
 - To be able to determine the thickness of the baseplate and thus better interpret the tomograms and the 3D maps of the fibers.
- 14- Why are there no membranes highlighted in segmented tomograms. This is very important information for fiber location and attachment.
- 15- Is it possible to estimate the length of the tube that protrudes from the baseplate in the contracted tails (and thus to estimate the maximum thickness of the membrane that can be crossed; and thus the maximum angle of inclination of the capsid to the membrane).
- 16- For the tail sheath : alphafold2 has been used to generate a model; for the inner tube; the T4 counterpart has been used. Why was AlphaFold2 not used?
- 17- Is it possible to isolate single fiber in the tomograms to analyze the hyperbranching mode?

Minor comments

Line 138: the second "and" has to be removed

Line 240: Fig 3A: shouldn't it be Fig 2A?

ml \diamond mL

microl \diamond microL

Reference 23 (line 796): the is incomplete (page number; volume number etc)

Reviewer #2 (Remarks to the Author):

Ouyang et al provide a mammoth description of the structure and aspects of the function of the *Klebsiella jumbo* myophage ϕ Kp24. The authors use traditional cryoEM approaches to determine the structure of the capsid and tail (with modelling aided by the structure prediction program AlphaFold2 and previously determined structural homologs), and then carry out tomographic studies of the virus infection processes aided by a machine learning algorithm that characterizes the different modes of fiber interaction with the bacterial cell and together, again, with AlphaFold2 attempt to predict the various depolymerase activities associated with the different types of fibers.

The capsid reconstruction is based on a relatively few numbers of particles (19,200) by modern standards and there is no mention of correcting for the Ewald sphere curvature distortions, although this may have been done implicitly in the processing programs. For particles this size this should certainly be discussed. If this was not applied, the resolution might be significantly improved with its use. At 4.3Å resolution there is certainly ambiguity with respect to the model and while the regions of the density shown with the model are respectable, there are no global criteria stated that adds confidence to the modelling. The description of the differences between the HK97 subunit and the much larger subunit in this structure would be easier to follow with a 2d secondary structure diagram (topology) with the changes inserted and highlighted with sequence numbers.

SEE UPLOADED FIGURES AS AN EXAMPLE

Following the discussion in the text is challenging. Likewise, the points of subunit interactions would be easier to understand if they were placed on a diagram like this.

Figure 1 will be improved if the authors have figure 1A and 1B in the same orientation, viewed down the icosahedral 3-fold axis. It would also be helpful if a hexamer were outlined in Figure 1A since these are not clear in the density depiction.

The tail structure was reconstructed segments from 400Å long segments of extended tails attached to full particle. In the text there is no discussion of the details of the procedure, just that it was performed with the Relion. There are more details in the extended information, but it would be useful to have a bit more detail in the text. The model building appears dependent on ALPHA-fold predictions as was the capsid protein. Descriptions of the tail protein structure would also benefit from a 2d comparison with outer sheath homolog ϕ KZ tail sheath protein gp29PR and the central sheath with rod-like (R)-type pyocin sheath protein (PDB ID: 6PYT). This would illustrate the similarities and differences in a readily understandable manner. Likewise, highlighting regions of interaction between the sheaths in the 2d depiction would nicely complement the 3d figures. Is it novel for these two domains to be fused together in a single polypeptide? If so, that should be emphasized

I am at a loss to review the last section of the paper since the approaches described employing machine learning are new to me. My uninformed reaction is not positive, however, since there does not appear to be enough information for a knowledgeable person to interpret the figures and follow the derivation of the resultant structures associated with the functions of the different fibers. As suggested below, I believe that this should be a separate paper.

Below are the criteria for the above general comments. I put a section from the paper in quotes to make them easier to find.

Page 4 What does this mean? Do they become a tailspike? "RBPs can adopt a tail fiber or tailspike shape."

Page 5 The authors should define jumbo phage "The recently described vB_KpM_FBKp24 (ϕ Kp24) jumbo myophage"

The following sentence fragment may apply to jumbo phage, but not to phage such as HK97 "most notably the composition of the entire capsid by a single MCP, and the presence of a tube in the center of the hexagons."

Page 9 there is an extra and in this sentence. "full capsid reconstructions and superimpose well"

Again, this may apply to jumbo phages, but not phages in general. "Unlike other capsids, the hexamers and pentamers are made up of copies of the same major capsid protein (MCP, gp372, accession no QQV92002)."

Page 10 This is where the 2D representation suggested would be most helpful. Is this information based totally on the structure prediction or was cryoEM data used to validate it? It appears to be based only on alphafold since the next paragraph describes the refinement against data. "In contrast, there are five anti-parallel β -sheets and four helices of different length in the corresponding domain of gp372 (C-terminal domain); (2) the N-terminal arm of HK97 gp5 monomer is in a mostly unstructured conformation, while the N-terminal of gp372 folds as a long helix (Extended Data Fig. 3)."

There should be a little more information in the body of the text as to what the procedure was for doing this. "To further refine the gp372 structure according to our 3D cryo-EM data and to determine putative interactions between MCPs in the native capsid assembly, we next reconstructed models of the capsid hexamer and pentamer assemblies, including the nearest neighboring hexamers (Extended Data Fig. 4)."

Page 11 I assume that the quality of the fit to the density was sufficient to validate the very detailed description based on the previous page. "Although the map resolution is not sufficient for accurate placement of all residue sidechains, ..."

Page 12 Do helices form the tube? Can you tell from the density or does alphafold indicate this? "Six of these long loops twist together to form the pore in the center of a hexamer, each loop extending from C-terminal domains of different gp372."

Page 15 There should be more detail in the main text about this reconstruction. Were the 400Å long helices divided into segments used for the reconstruction or were the entire helices treated as a rigid unit. How many helices were used? "For the reconstruction of the tail tube and sheath, we picked 400 Å long segments from phages that contained full capsids (Extended Data Fig. 5B). After 3D refinement, the tail structure (EMD-14357, Fig. 3A) in its extended conformation was determined to a resolution of 4.1 Å in Relion14 (Extended Data Fig. 4C)".

Page 16 Is the phiKZ also a two domain protein with a pyocin domain? There should be some comment about the novelty of the two domain structure in gp118 or not. "It is a homologue of the bacteriophage phiKZ tail sheath protein gp29PR (PDB ID: 3JOH)."

Page 18 What is the basis for the homology? As in many other places in the paper it is not clear how much of the model discussed is based on current data, alphafold or the structures of other homologs. "The tube of phage ϕ Kp24 is homologous to the tube of other contractile injection systems, such as phage T428, R-type pyocins²⁷, *Serratia entomophila* antifeeding prophage (afp)²⁹, and the bacterial Type VI Secretion System (T6SS)^{30,31}"

Page 22 Should be a "we" inserted I believe. "rounds of optimization, applied the trained network"

Some comment should be made as to the criteria that these are meaningful. "After extraction and format conversion, 89 tail fiber structures of pre-infection phage and 338 tail fiber structures of post-infection attached phage were generated. 40 representative tail fiber structures of pre-infection and tail fiber structures of post-infection are shown in Extended Data Figs. 6 and 7. "

Pages 22-27. I think that this should be a separate paper. It is a technical tour de force that is far from intuitive to a typical structural virologist. It needs to be expanded in terms of methods and interpretation and detracts from the rest of the paper in its current form.

Reviewer #3 (Remarks to the Author):

The manuscript by Ouyang et al. describes the structure and structural rearrangements of the myophage phi-Kp24, which infects *Klebsiella pneumoniae*. *K. pneumoniae* is a major pathogen and an important target for antibiotics, however, antimicrobial resistance is an increasing problem. The development of phage therapy could be a promising strategy against the resistance problem. The present work is an important step towards this goal, as it describes in great detail the structure of the entire phage, i.e. capsid, tail and baseplate including cell surface interactions using a powerful combination of several techniques (cryo-EM and -ET, MD, structure prediction, and machine learning). The phi-Kp24 phage is also particularly interesting as it seems to have a broad infectivity for different bacterial strains, which was also tested experimentally in this work. The phage phi-Kp24 interestingly shows heavily branched tail fibers, which is very unusual. These fibers rearrange substantially upon cell surface engagement (studied here by cryo-ET).

Overall this is very interesting work, the manuscript is very well written and the figures are clear. There are only a few minor issues that from my point of view should be addressed:

1) There is an unusual tube in the center of the hexagons in the capsid. The authors speculate in the Discussion what the purpose of the pore could be. A few more details in the Results section would be helpful, though. There seems to be density in the pore, could this be an ion? Or is the density much larger? What is the size of the pore/size of the density blob inside the pore? And what about the properties of the pore, is it hydrophobic/polar/charged?

2) The resolution of the capsid (4.3Å) and of the tail (4.1Å) is apparently not sufficient to build an atomic model de novo. The authors used models of gp372 and gp118 predicted by AlphaFold2 and another PDB entry (5w5f) as a model for the inner tube, and refined them into the density map using ISOLDE and MDFF, which is a very reasonable approach. Several conclusions are drawn from these refined atomic models, including details of subunit interactions. It would be good to provide some more detail on the quality of these models. AlphaFold2 yields a predicted IDDT, which would be good to have in the Supplement. Also, how much have the models changed during the refinement (in RMSD, GDT or TMscore) from the initial models (AlphaFold2 prediction/PDB template) to the final refined model?

In addition, it would be good to show model-map FSC curves and determine the cross-resolution.

3) line 330: The model for tail assembly. This section is written as if the order of assembly events has been clearly determined. The authors should either provide more evidence that the assembly must happen as described, or write the text in a way that makes it clear to the reader that this is an hypothesis ("We propose/suggest...").

typos:

line 138: "... reconstructions *and* superimpose well" remove *and*

line 278: "...negative*ly* charged patch..."

List of Responses

Dear Editors and Reviewers:

Thank for your constructive comments concerning our manuscript entitled "High resolution reconstruction of a Jumbo bacteriophage infecting capsulated bacteria using hyperbranched tail fibers". We have addressed each input as outlined below, which has improved the manuscript significantly. The revised text is marked red in the manuscript to highlight the changes we've made. The main corrections in the manuscript and the responses to the reviewers' comments are outlined below.

Responds to the reviewers' comments:

Reviewer 1:

The paper "High resolution reconstruction of a Jumbo bacteriophage infecting capsulated bacteria using hyperbranched tail fibers" by Ouyang et al. describes the structural analysis of the jumbo myophage ϕ Kp24. The structure of the head of this phage, resolved at 4.3 Å (the highest up to now deposited on the EMDB database) reveals a triangulation number $T=27$ and that this structure is based on a single major capsid protein. The structure of the tail of this phage was also solved at 4.1 Å which also allowed to generate a quasi-atomic model of this part. Finally, this work focused on the hyperbranched fibers of this phage. An electron tomography approach coupled with machine learning methods was used to show dramatic rearrangement upon attachment to the cell surface.

If several points of this manuscript are very interesting, new and the approach very elegant (the analysis of the fiber in particular), several technical points deserve to be

improved and explained in more detail before publication:

We thank the reviewer for the kind remarks about our manuscript and for the useful comments, which we address in detail below.

Major points:

1. Almost no mention is made of previous structural work (even at low resolution) on jumbo phages and in particular phages with triangulation number $T=27$ (RSL1, RSL2, phiKZ). The same is true for phages showing multiple fibers on the baseplate or tail.

Response: Thank you for this comment. We included additional information about jumbo phage capsids and phages with multiple fibers as follows:

“The capsid of *Pseudomonas aeruginosa* Jumbo phage phiKZ¹⁵, and *Ralstonia solanacearum* phages ϕ RSL1¹⁶, and ϕ RSL2¹⁷, are built in the HK97-fold^{18,19} with a triangulation number of $T = 27$. Similar to these phage capsids, the capsid of ϕ Kp24 is built up by a lattice of hexamers and pentamers in each facet (Fig. 1B). The hexamers and pentamers of ϕ Kp24 are made up of copies of the same major capsid protein (MCP, gp372, accession number QQV92002).” (Page 9, line 140 to 145)

“Phages from Menlow group⁸ (KpS110 and 0507-KN2-1), and phage Φ K64-1 group⁶ (Φ K64-1 and RaK2) are the *Klebsiella* phages with the most elaborate tail fiber apparatus described so far, comprising between five and eleven tail fibers.” (Page 28, line 464, 466)

2. The general EM table is missing (Data collection; Refinement; Model composition; RMS deviations, validation, Ramachandran plot).

Response: Thank you for pointing this out. We have added the data collection and

refinement parameters in the Extended Table 1 (two capsids), and Extended Table 3 (extended tail).

The validation, RMSD, Ramachandran plot, Rama-Z, and Model-Data CC data have been added:

Two capsids: Extended Table 2, Extended data Fig. 7, and Extended Fig. 8.

Outer sheath and inner tube: Extended Table 4, Extended data Fig. 14, and Extended Fig. 20.

3. It is written: line 980: The full capsid (cyan) is a little smaller than empty capsid (gray). What does “little” mean?

Response: We have added the following text to address this question:

“The diameter of the full capsid is about 8 Å shorter than the diameter of the empty capsid from vertex to vertex, and about 12 Å shorter along the 2-fold symmetry axis. The DNA is tightly packed inside the full capsid. Additional density is present inside the capsid under the five-fold vertices. However, the major capsid proteins are insufficient in size to account for these densities, and therefore we hypothesize that they arise from internal decoration.” (Page 55, line 1026 to 1030)

4. The phage capsid is large and therefore the defocus difference between the top of the capsid and the bottom of the capsid in the ice is important. Have you tried to perform correction of the Ewald sphere during image processing?

Response: Thank you very much for your insightful comments. Following your advice, we ran the Ewald sphere correction on the capsids using a single side-band image processing algorithm after 3D refinement. It improved the full capsid from 4.7 Å to 4.3

Å and the empty capsid from 4.3 Å to 4.1 Å, which is the highest resolution we can expect of the particles (the pixel size is 2.055 Å). These improved maps show more details of the capsid structure and we have re-refined our models accordingly. Furthermore, we have unsuccessfully tried to use un-binned data (raw pixel size is 1.37 Å). The large diameter of the capsids (~1500 Å) combined with the available processing infrastructure requires data binning by 1.5.

5. Comparison with HK97 MCP: is there evidence of cross-linking? The hypothesis is not even mentioned? Please add a comment on that.

Response: We apologize for this confusion. While the proteins show some structural homology, BLAST shows no similarity in sequence between HK97 MCP and gp372. While no similarity is found at sequence level, the structural homology suggests that cross-linking of MCP proteins may occur similarly to that described for HK97 MCP.

6. Has PISA been used to see/ validate the aa interactions?

Response: Thank you for this suggestion. Given that our maps do not generally provide information on side chain conformation and our analysis was confined to potential salt bridges. This approach was suitable for isolating potential contacts shared between localized regions of analogous MCP-MCP interfaces. However, we didn't use PISA to validate the interactions.

7. Is reference 18 the correct one? I don't think that the number of MCP is mentioned in the reference.

Response: Thank you for pointing this out. We have deleted the reference, and modified the text as follows:

“The capsid contains a total of 260 hexamers (13 per facet) and 11 pentamers (one vertex is occupied by the portal complex), which represent 1,615 copies of the major capsid protein (MCP).” (Page 9, line 145 to 147)

8. Comparison between the hexamers: please provide RMSD

Response: Thank you, we now report RMSD values for these comparisons in the main text:

“The overall conformations of the central and surrounding hexamers are quite similar, displaying a backbone root-mean-square displacement (RMSD) of around 1.5 Å.” (Page 12, line 197 to 198)

“Comparison of the hexamers surrounding the pentamer with those from the hexameric assembly show that they are also slightly more curved and display an RMSD of 3-5 Å.” (Page 14, line 231 to 233)

9. Figure 2B: the superimposition of the hexamer MCP and the pentamer MCP have been performed on the arm part. Is it possible to also do it on the triangular body?

Response: We have added the superimposition to Fig. 2 and modified the text as follows:

“There are some differences, such as the C-terminal loop, the N-terminal α helix, and the pendulum angle (upper red arrow) of two triangular bodies (aligned hand and arm), or the pendulum angle (lower red arrow) of two hands (aligned triangular body).” (Page 13, 213 to 215).

10. Tail: it seems that the picking was done in adjacent boxes. Normally there can be an overlap between boxes (it will increase the number of boxes and the final resolution)

and moreover by imposing the helical symmetry and a C6 symmetry this would greatly improve the resolution. I suggest that this be done. This would increase the resolution (better than 3.5Å) and directly enable building of the chain.

Response: We have re-run a helix reconstruction of the tail, added C6 symmetry, and determined the helix parameters. As a result, we improved the resolution to 3.0 Å in the extended tail structure. Subsequently, we rebuilt the model of the sheath and tube. (Helical reconstruction workflow and helix parameter table, Page 67, 68, 69)

11. What are the helical parameters of the tail?

Response: We have added these parameters in the main text (Page 16, line 274 to 275), helical reconstruction workflow and helix parameter table in extended data (page 67, 68, 69) and described more details about reconstruction in Methods (Page 40).

Page 16, line 274 to 275: “Using HI3D, the helical rise and twist are determined at 39.03 Å and 20.89°, respectively.”

Page 40, line 720 to 722: “The helical rise and twist parameters for final 3D refinement were analyzed using HI3D from the Jiang lab (Extended table. 3), the helical rise and twist are determined at 39.03 Å and 20.89°, respectively.”

12. Since there are empty capsids, there are certainly contracted tails. Why was no analysis done on the contracted tails? It could be nice to compare the contracted and non-contracted tails.

Response: This would indeed be an interesting comparison. However, for our SPA data collection, we used a purified phage sample. This results in the sufficient concentration of capsids for our analysis, but it also means that the observed number of contracted

tails is too small for structural determination. We have tried to overcome this by helical-auto-picking using a template with contracted tail particles, but we only got a few thousand particles, with a majority of these particles being damaged. Therefore, we failed to get an acceptable reconstruction.

13. No image analysis was performed on the baseplate. Why?

This would allow 2 things:

- To see the origin of the fibers and maybe even solve the structure or part of the structure of some of them.
- To be able to determine the thickness of the baseplate and thus better interpret the tomograms and the 3D maps of the fibers.

Response: We agree with the reviewer that image analysis of the baseplate would be very interesting. Indeed, we have attempted this but were not successful so far. We assume that the reason for this is that baseplate is too obscured by the network of the tail fibers, preventing successful auto-picking of the particles. We also tried to manually pick particles of baseplate. With this approach, we could pick about two thousand particles, but this number was not sufficient for a good reconstruction. We aim to address this in the future by mutating the phage to lack the tail fibers, but this is out of the scope of this work.

14. Why are there no membranes highlighted in segmented tomograms. This is very important information for fiber location and attachment.

Response: Thank you for this suggestion. We have not done this due to the missing wedge: Most of the post-infection phages are located at the top of lysed cells, where

the membranes are oriented parallel to the grid. Therefore, we couldn't mark/highlight the surface of the membrane with any accuracy. However, we feel that the structural comparison between the attached and unattached tail fiber architectures are meaningful even if the membranes are not segmented. In the future, this limitation may be overcome or minimized by dual axis tomography, but these experiments are out of scope for this initial analysis.

15. Is it possible to estimate the length of the tube that protrudes from the baseplate in the contracted tails (and thus to estimate the maximum thickness of the membrane that can be crossed; and thus the maximum angle of inclination of the capsid to the membrane).

Response: We have now measured hundreds of inner tubes from the top tip of the collar connection (capsid side) to the end tip of the spike side. The length of the tube is about 180 nm on average. We have included this length measurement in the text (Page 16, line 267 to 268). The thickness of bacteria cell wall is about 30 nm, and the tube length is indeed enough to cross the membrane. However, it is hard to measure the angle of inclination. Because of the missing wedge, we couldn't accurately determine the plane of membrane. In addition, most of the tomograms show the top view of bacteria, where the membrane is parallel to the grid's surface and hard to segment. We selected some phages in a side-view, and tried to measure the angle of inclination during infection in an easy way (showed below). From limited number of measurements, we determined an angular range between 23° to 82° . However, these angles are not very accurate, so we are hesitant to estimate a maximum angle of inclination of the capsid to the membrane.

16. For the tail sheath: alphafold2 has been used to generate a model; for the inner tube; the T4 counterpart has been used. Why was AlphaFold2 not used?

Response: We have run HMMER on all Kp24 phage proteins to find the inner tube protein, and subsequently used alphafold2 to generate a model for this protein (gp119). This model was used to fit into the improved 3.0 Å density map of the inner tube, as shown in the updated fig. 4.

17. Is it possible to isolate single fiber in the tomograms to analyze the hyperbranching mode?

Response: We have tried to do this but were not successful. The large number of tail fibers and their seemingly unordered arrangement, together with the low signal to noise ratio and the missing wedge artifact prevented the tracing of individual fiber (even in denoised datasets). In the future, we hope to tackle this problem, but it will involve

developing suitable isolation methods and the generation of genetic mutants, which is out of scope of the current study.

Minor comments

Line 138: the second “and” has to be removed. **Fixed**

Line 240: Fig 3A: shouldn't it be Fig 2A? **Fixed, it should be Fig. 2A.**

ml \square mL **Fixed**

microl \square microL **Fixed**

Reference 23 (line 796): the is incomplete (page number; volume number etc) **Fixed**

Reviewer 2:

Ouyang et al provide a mammoth description of the structure and aspects of the function of the Klebsiella jumbo myophage ϕ Kp24. The authors use traditional cryoEM approaches to determine the structure of the capsid and tail (with modelling aided by the structure prediction program AlphaFold2 and previously determined structural homologs), and then carry out tomographic studies of the virus infection processes aided by a machine learning algorithm that characterizes the different modes of fiber interaction with the bacterial cell and together, again, with AlphaFold2 attempt to predict the various depolymerase activities associated with the different types of fibers.

We thank the reviewer for the positive comments on our manuscript and for the constructive criticism provided, which we address in detail below.

The capsid reconstruction is based on a relatively few numbers of particles (19,200) by modern standards and there is no mention of correcting for the Ewald sphere curvature distortions, although this may have been done implicitly in the processing programs. For particles this size this should certainly be discussed.

Response: As suggested also by Reviewer 1, we ran the Ewald sphere correction on the capsids, which improved resolution of the full capsid from 4.7 Å to 4.3 Å and the empty capsid from 4.3 Å to 4.1 Å, which is the highest resolution we can expect of the particles (the pixel size is 2.055 Å). These improved maps show more details of the capsid structure.

If this was not applied, the resolution might be significantly improved with its use. At 4.3Å resolution there is certainly ambiguity with respect to the model and while the regions of the density shown with the model are respectable, there are no global criteria stated that adds confidence to the modelling. The description of the differences between the HK97 subunit and the much larger subunit in this structure would be easier to follow with a 2d secondary structure diagram (topology) with the changes inserted and highlighted with sequence numbers.

Thank you for this suggestion. Unfortunately, this is not possible because we cannot align the two sequences.

Following the discussion in the text is challenging. Likewise, the points of subunit interactions would be easier to understand if they were placed on a diagram like this.

Response: Thank you for this suggestion. We generated a 2D secondary structure diagram comparing the HK97's gp5 and Kp24 subunits, using the example provided. This is now shown as Extended data fig. 4, 5, and 6 (Page 59, 60, 61).

1. Figure 1 will be improved if the authors have figure 1A and 1B in the same orientation, viewed down the icosahedral 3-fold axis. It would also be helpful if a hexamer were outlined in Figure 1A since these are not clear in the density depiction.

Response: Thank you for your suggestions. We have updated Fig. 1A and Fig. 1B to be in the same orientation, viewed down the icosahedral 3-fold axis. (Page 7, Fig. 1)

2. The tail structure was reconstructed segments from 400Å long segments of extended tails attached to full particle. In the text there is no discussion of the details of the procedure, just that it was performed with the Relion. There are more details in the extended information, but it would be useful to have a bit more detail in the text. The model building appears dependent on ALPHA-fold predictions as was the capsid protein. Descriptions of the tail protein structure would also benefit from a 2d comparison with outer sheath homolog phiKZ 3J0H tail sheath protein gp29PR and the central sheath with rod-like (R)-type pyocin sheath protein (PDB ID: 6PYT). This would illustrate the similarities and differences in a readily understandable manner. Likewise, highlighting regions of interaction between the sheaths in the 2d depiction would nicely complement the 3d figures. Is it novel for these two domains to be fused together in a single polypeptide? If so, that should be emphasized

Response: We have now re-run a helix reconstruction of the extended tail. Furthermore, we have now added helix parameters (Page 16, line 274 to 275), the helical reconstruction workflow and the helix parameter table in extended data (Page 67,68,69). Additionally, we have added more details about reconstruction in the Methods. (Page 40)

Page 16, line 274 to 275: “Using HI3D, the helical rise and twist are determined at 39.03 Å and 20.89°, respectively.”

Page 40, line 720 to 722: “The helical rise and twist parameters for final 3D refinement were analyzed using HI3D from the Jiang lab (Extended table. 3), the helical rise and twist are determined at 39.03 Å and 20.89°, respectively.”

Using ENDscript 3, the BLAST result shows that only the outer sheath component is a homologue of the tail sheath protein gp29PR (PDB ID: 3SPE) from bacteriophage phiKZ. There is no sequence similarity of the central sheath component. We have added a 2D comparison of the tail sheath protein gp29PR from bacteriophage phiKZ (PDB ID: 3SPE) and the outer sheath component of Kp24 (Extended Data Fig. 15, Page 73).

The regions of interaction between the sheaths are highlighted (N-terminal extension, red; C-terminal domain, orange; C-terminal extension, green) in the extended data Fig. 15, (Page 73).

3. I am at a loss to review the last section of the paper since the approaches described employing machine learning are new to me. My uninformed reaction is not positive, however, since there does not appear to be enough information for a knowledgeable person to interpret the figures and follow the derivation of the resultant structures associated with the functions of the different fibers. As suggested below, I believe that this should be a separate paper.

Response: Thank you for your suggestion. However, we feel that the analysis of the structural organization of the tail fibers into a hyperbranched system is very important, since it is unique in comparison to most other published phage tail architectures. We note that the other two reviewers share our view that the machine learning analysis of the fiber structures is an important aspect of the paper, with Reviewer 1 specifically calling “the approach very elegant (the analysis of the fiber in particular)” and Reviewer

3 including machine learning in the “powerful combination of several techniques” employed.

However, we do fully agree with the reviewer that it is important that the machine learning parts are clearly explained in the manuscript, especially for readers that might not be familiar with these techniques. In the revised version, we have reworked and shortened the parts related to machine learning to improve their clarity. We have also included additional references for this approach:

Pelt, D.M., Batenburg, K.J. & Sethian, J.A. Improving tomographic reconstruction from limited data using mixed-scale dense convolutional neural networks. *Journal of Imaging* 4, 128 (2018).

Segev-Zarko, L.-a. et al. Cryo-electron tomography with mixed-scale dense neural networks reveals key steps in deployment of *Toxoplasma* invasion machinery. *PNAS Nexus* (2022).

Below are the criteria for the above general comments. I put a section from the paper in quotes to make them easier to find.

4. Page 4 What does this mean? Do they become a tailspike? “RBPs can adopt a tail fiber or tailspike shape. “

Response: We apologize for the unclear description. We have rewritten this paragraph on page 4 as follows:

“Even though multidrug-resistant *K. pneumoniae* are insensitive to standard-of-care antibiotics, they remain susceptible to bacteriophage infection. Bacteriophages, or phages for short, are viruses that infect bacteria. *Klebsiella*-specific phages can successfully infect and kill their natural host; however, most of these phages are typically highly strain-specific due to the variable capsular polysaccharides (CPS) of this species, which act as a primary phage receptor. Capsule-dependent *Klebsiella*

phages, including phages Φ K64-1 or vB_KleM-RaK26,7, are equipped with tail fibers or tailspikes serving as receptor-binding proteins (RBPs) containing CPS-degrading enzyme domains (coined capsule depolymerases) that enable successful phage adsorption and infection⁸. For simplicity, we henceforth use the term ‘tail fiber’.”

5. Page 5 The authors should define jumbo phage “The recently described vB_KpM_FBKp24 (ϕ Kp24) jumbo myophage “

Response: We had added some sentences to define and explain the jumbo phage: “Tailed bacteriophages with genomes of more than 200 kbp of DNA are defined as jumbo phages⁹. The most notable structural features of jumbo phages are large capsids that encapsulate their genome¹⁰.” (Page 5, line 67 - 69)

6.The following sentence fragment may apply to jumbo phage, but not to phage such as HK97 “most notably the composition of the entire capsid by a single MCP, and the presence of a tube in the center of the hexagons.”

Response: Thank you for pointing this out. We have modified this sentence as follows:

“The data revealed unusual features of the capsid of ϕ Kp24, most notably the composition of the entire capsid by a single MCP, and the presence of a pore in the center of the hexagons.” (Page 5, line 81 to 83).

7. Page 9 there is an extra and in this sentence. “full capsid reconstructions and superimpose well”

Response: We have corrected this error. (Page 9, line 135)

“Since the empty and full capsid reconstructions superimpose well”

8. Again, this may apply to jumbo phages, but not phages in general. “Unlike other capsids, the hexamers and pentamers are made up of copies of the same major capsid protein (MCP, gp372, accession no QQV92002).”

Response: We agree with the reviewer that the description indeed does not apply to phages in general. We have therefore adjusted the text as follows:

“Similar to these phage capsids, the capsid of ϕ Kp24 is built up by a lattice of hexamers and pentamers in each facet (Fig. 1B). The hexamers and pentamers of ϕ Kp24 are made up of copies of the same major capsid protein (MCP, gp372, accession number QQV92002).” (Page 9, line 142 to 145)

9. Page 10 This is where the 2D representation suggested would be most helpful. Is this information based totally on the structure prediction or was cryoEM data used to validate it? It appears to be based only on AlphaFold since the next paragraph describes the refinement against data. “In contrast, there are five anti-parallel β -sheets and four helices of different length in the corresponding domain of gp372 (C-terminal domain); (2) the N-terminal arm of HK97 gp5 monomer is in a mostly unstructured conformation, while the N-terminal of gp372 folds as a long helix.”

Response: Thank you for your suggestion. The BLAST result showed that there is no similarity in sequence between HK97 MCP gp5 and gp372. However, we have added the 2D secondary structure diagrams of gp372 and HK97's gp5 (Extended data Fig. 5, page 60; and Extended data Fig. 6, page 61). The secondary structure information for gp372 is based on the AlphaFold2 prediction. Although our flexible fitting simulations of gp372 make use of secondary structure restraints to prevent overfitting, we achieve an excellent overlap with the map as demonstrated the model validation metrics provided in Extended Table 2.

10. There should be a little more information in the body of the text as to what the procedure was for doing this. “To further refine the gp372 structure according to our 3D cryo-EM data and to determine putative interactions between MCPs in the native capsid assembly, we next reconstructed models of the capsid hexamer and pentamer assemblies, including the nearest neighboring hexamers (Extended Data Fig. 4).”

Response: Thank you for highlighting this need for additional detail. We have added a brief description of the modelling procedure in the main text and pointed the reader to the associated Methods section where they can find full detail.

“Briefly, models of the hexamer and pentamer assemblies were first constructed via rigid docking of the predicted gp372 structure and subsequently refined to our 4.1 Å map using molecular dynamics flexible fitting (MDFF). The resulting hexamer model was then rigidly docked to the corresponding regions surrounding the refined pentamer and hexamer assemblies, and the combined structures were subjected to an additional MDFF simulation. The obtained models therefore provide representative information on each unique MCP-MCP interaction within the capsid, allowing for an assessment of key residue-residue interactions likely crucial for capsid stability.” (Page 11, line 175 to 183)

11. Page 11 I assume that the quality of the fit to the density was sufficient to validate the very detailed description based on the previous page. “Although the map resolution is not sufficient for accurate placement of all residue sidechains, ...”

Response: We agree with the reviewer and have deleted the sentence: “Although the map resolution is not sufficient for accurate placement of all residue side chains, we can still observe prominent densities throughout the gp372 structure corresponding to likely side chain conformations.”

12. Page 12 Do helices form the tube? Can you tell from the density or does alphafold indicate this? “Six of these long loops twist together to form the pore in the center of a hexamer, each loop extending from C-terminal domains of different gp372.”

Response: Yes, the central short pore of the hexamer is indeed helical according to the figure below. This becomes more obvious if we apply a smaller threshold (see picture below). We were able to flexibly fit six long loops of gp372s into the corresponding density map. Memory constraints prevent the folding of multiple gp372s using AlphaFold2, so the association of neighboring C-terminal loops is not determined. Nevertheless, our flexible fitting simulations of six gp372 proteins organized as a hexamer readily form the helical arrangement suggested by the density.

13. Page 15 There should be more detail in the main text about this reconstruction. Were the 400Å long helices divided into segments used for the reconstruction or were the entire helices treated as a rigid unit. How many helices were used? “For the reconstruction of the tail tube and sheath, we picked 400 Å long segments from phages that contained full capsids (Extended Data Fig. 5B). After 3D refinement, the tail structure (EMD-14357, Fig. 3A) in its extended conformation was determined to a resolution of 4.1 Å in Relion14 (Extended Data Fig. 4C)”.

Response: Following your suggestion, we have re-run a helix reconstruction of the extended tail, added C6 symmetry, and determined the helix parameters. This improved

the extended tail structure to 3.0 Å resolution, and allowed us to rebuild the model of the sheath. We have added the helix parameters in the main text (Page 16, line 274 to 275), the helical reconstruction workflow and helix parameter table in extended data (Page,67,68,69), and described more details about the reconstruction in the Methods. (Page 39 to 40).

14. Page 16 Is the phiKZ also a two domain protein with a pyocin domain? There should be some comment about the novelty of the two domain structure in gp118 or not. “It is a homologue of the bacteriophage phiKZ tail sheath protein gp29PR (PDB ID: 3JOH).”

Response: We apologize for the unclear description. Only the outer sheath component is a homologue of the tail sheath protein gp29PR (PDB ID: 3SPE) from bacteriophage phiKZ. We have adjusted the text to clarify this:

“The sequence analysis by ENDscript3 shows that the outer sheath component is a homologue of the tail sheath protein gp29PR (PDB ID: 3SPE) from bacteriophage phiKZ (Extended Data Fig. 15).” (Page 16, line 281 to 283)

15. Page 18 What is the basis for the homology? As in many other places in the paper it is not clear how much of the model discussed is based on current data, alphafold or the structures of other homologs. “The tube of phage φKp24 is homologous to the tube of other contractile injection systems, such as phage T428, R-type pyocins²⁷, Serratia entomophila antifeeding prophage (afp)²⁹, and the bacterial Type VI Secretion System (T6SS)^{30,31}”

Response: We identified the inner tube protein (gp119) of phage Kp24 using HMMER, and this protein shares 61.5% similarity with the baseplate wedge protein of Serratia phage Moabite, and 36.3% similarity with the putative tail tube protein of Pseudomonas

phage pPa_SNUABM_DT01. We generated a 3D model of gp119 of phage Kp24 using alphafold2 and used this model to fit into the density map of the inner tube. Due to these changes, we modified the text as follows: (page 19)

“The tube of other contractile injection systems, such as phage T4³⁰, R-type pyocins³¹, *Serratia entomophila* antifeeding prophage (afp)³², and the bacterial Type VI Secretion System (T6SS)^{33,34}, show the ability to translocate different substrates and the nature of the substrate determines the properties of the tube’s channel³⁰. The tube of phage ϕ Kp24 shows a similar function.

The structure of the tube protein (gp119, accession number QQV92088.1, Fig. 4A) was predicted using AlphaFold2 and flexible fitted into the ϕ Kp24 tail EM map. We used gp119 to build a model of the tube of phage ϕ Kp24. One ring-like structure consisting of six tube proteins represents an assembly unit of the inner tube of phage ϕ Kp24 (Figs. 4B and 4C)

16. Page 22 Should be a “we” inserted I believe. “rounds of optimization, applied the trained network”

Response: We have adjusted the text, line 402, page 23:

“after several rounds of optimization, we applied the trained network to all tomograms.”

17. Some comment should be made as to the criteria that these are meaningful. “After extraction and format conversion, 89 tail fiber structures of pre-infection phage and 338 tail fiber structures of post-infection attached phage were generated. 40 representative tail fiber structures of pre-infection and tail fiber structures of post-infection are shown in Extended Data Figs. 6 and 7. “

Response: Our purpose is to compare the conformational changes between pre-infection and post-infection phages. Therefore, we selected by visual inspection and showed 40 representative tail fiber structures each of pre-infection and post-infection phages in Extended data. We also defined the pre/post infection phages and added the text of as follows:

“We define the pre-infection phages as follows: (1) the capsid is full, (2) the tail is extended, (3) the phage is intact, and (4) the phage is not in contact with a bacterial cell. Similarly, we define the post-infection phages as follows: (1) the capsid is empty, (2) the tail is contracted, and (3) there are a few connections between the tips of fibers and the cell membrane.” (Page 23, line 403 to 407)

18. Pages 22-27. I think that this should be a separate paper. It is a technical tour de force that is far from intuitive to a typical structural virologist. It needs to be expanded in terms of methods and interpretation and detracts from the rest of the paper in its current form.

Response: Thank you for your suggestion. We agree that this is not typically part of a structural phage analysis. However, the reason for this is that this tail fiber architecture of ϕ Kp24 is not that of a typical phage, but a largely uncharacterized feature that has not been described previously. In addition, the tail fibers likely contribute to the ability of this phage to infect a variety of serotypes, which is highly important and makes it a promising candidate for future phage therapies. Therefore, we believe that this analysis is an essential part of the characterization of this phage and should remain as part of this study. We have reworked this section to make it more readable.

Reviewer 3:

The manuscript by Ouyang et al. describes the structure and structural rearrangements of the myophage phi-Kp24, which infects *Klebsiella pneumoniae*. *K. pneumoniae* is a major pathogen and an important target for antibiotics, however, antimicrobial resistance is an increasing problem. The development of phage therapy could be a promising strategy against the resistance problem. The present work is an important step towards this goal, as it describes in great detail the structure of the entire phage, i.e. capsid, tail and baseplate including cell surface interactions using a powerful combination of several techniques (cryo-EM and -ET, MD, structure prediction, and machine learning). The phi-Kp24 phage is also particularly interesting as it seems to have a broad infectivity for different bacterial strains, which was also tested experimentally in this work. The phage phi-Kp24 interestingly shows heavily branched tail fibers, which is very unusual. These fibers rearrange substantially upon cell surface engagement (studied here by cryo-ET).

Overall, this is very interesting work, the manuscript is very well written and the figures are clear. There are only a few minor issues that from my point of view should be addressed:

We thank the reviewer for the kind remarks about our manuscript and for the useful comments, which we address in detail below.

1. There is an unusual tube in the center of the hexagons in the capsid. The authors speculate in the Discussion what the purpose of the pore could be. A few more details in the Results section would be helpful, though. There seems to be density in the pore, could this be an ion? Or is the density much larger? What is the size of the pore/size of

the density blob inside the pore? And what about the properties of the pore, is it hydrophobic/polar/charged?

Response: We assume that the density in the center of the pore is noise or dust. Not every pore has this density blob in the center. The outer diameter of the pore is about 20 Å and the inner diameter is about 15 Å. The density blob has different sizes ranging from 1 Å to 3 Å. We tested the properties of the pore's inner surface, and this information was added to the main text:

“Analysis of the electrostatics along the pore (Extended Data Fig. 10) show that near the entrance (from the outside into the inner capsid) it is hydrophilic, while from the opposite side, it is lipophilic. Moreover, at near the pore center, it contains an annular section that is negatively charged.” (Page 12, line 203 to 206)

We also added our hypothesis in Discussion:

“However, pores have been reported in viruses, for example in HIV-1. This virion uses dynamic capsid pores to import nucleotides and fuel encapsulated DNA synthesis⁵¹. We speculate that the pore of ϕ Kp24 may be involved in balancing pressure differences during loading of the DNA into the capsid, or during DNA ejection when capsid size increases.” (Page 31, line 524 to 526)

2. The resolution of the capsid (4.3\AA) and of the tail (4.1\AA) is apparently not sufficient to build an atomic model de novo. The authors used models of gp372 and gp118 predicted by AlphaFold2 and another PDB entry (5w5f) as a model for the inner tube, and refined them into the density map using ISOLDE and MDFF, which is a very reasonable approach. Several conclusions are drawn from these refined atomic models, including details of subunit interactions. It would be good to provide some more detail on the quality of these models. AlphaFold2 yields a predicted IDDT, which would be good to have in the Supplement. Also, how much have the models changed during the refinement (in RMSD, GDT or TMscore) from the initial models (AlphaFold2 prediction/PDB template) to the final refined model? In addition, it would be good to show model-map FSC curves and determine the cross-resolution.

Gp372

Gp118

Gp119

We have added the data collection and refinement parameters in the Extended Table 1 (two capsids), and Extended Table 3 (extended tail).

The validation, RMSD, Ramachandran plot, Rama-Z, and Model-Data CC data have been added:

Two capsids: Extended Table 2, Extended data Fig. 7, and Extended Fig. 8.

Outer sheath and inner tube: Extended Table 4, Extended data Fig. 14, and Extended Fig. 20.

3. line 330: The model for tail assembly. This section is written as if the order of assembly events has been clearly determined. The authors should either provide more evidence that the assembly must happen as described, or write the text in a way that makes it clear to the reader that this is a hypothesis ("We propose/suggest...").

Response: We agree with the reviewer: we could not define the order of the tail assembly of our phage according to the cryo-EM data. Therefore, we have adjusted the text:

“Similar to other phage-tail systems, we propose that” (Line 350, page 20).

typos:

line 138: "... reconstructions *and* superimpose well" remove *and* fixed

line 278: "...negative*ly* charged patch..." fixed

REVIEWERS' COMMENTS

Reviewer #1 (Remarks to the Author):

I am now happy with the new version of the manuscript. Thanks

Reviewer #2 (Remarks to the Author):

I have read the response to the reviewers carefully and the manuscript. I feel that the authors have done a credible job to address the numerous comments from the reviewers. I have no further recommendations

Reviewer #3 (Remarks to the Author):

The authors have addressed all my concerns convincingly.

Reviewers' comments

Reviewer 1 (Remarks to the Author):

I am now happy with the new version of the manuscript. Thanks

Thank you.

Reviewer 2 (Remarks to the Author):

I have read the response to the reviewers carefully and the manuscript. I feel that the authors have done a credible job to address the numerous comments from the reviewers.

I have no further recommendations

Thank you.

Reviewer 3 (Remarks to the Author):

The authors have addressed all my concerns convincingly.

Thank you.